# Exploring Low-Dimensional Subspaces in Diffusion Models for Controllable Image Editing

**Siyi Chen**[1*]    **Huijie Zhang**[1*]    **Minzhe Guo**[1]    **Yifu Lu**[1]    **Peng Wang**[1]    **Qing Qu**[1]

[1]University of Michigan

{siyich,huijiezh,vincegmz,yifulu,pengwa,qingqu}@umich.edu

## Abstract

Recently, diffusion models have emerged as a powerful class of generative models. Despite their success, there is still limited understanding of their semantic spaces. This makes it challenging to achieve precise and disentangled image generation without additional training, especially in an unsupervised way. In this work, we improve the understanding of their semantic spaces from intriguing observations: among a certain range of noise levels, (1) the learned posterior mean predictor (PMP) in the diffusion model is locally linear, and (2) the singular vectors of its Jacobian lie in low-dimensional semantic subspaces. We provide a solid theoretical basis to justify the linearity and low-rankness in the PMP. These insights allow us to propose an unsupervised, single-step, training-free **LO**w-rank **CO**ntrollable image editing (LOCO Edit) method for precise local editing in diffusion models. LOCO Edit identified editing directions with nice properties: homogeneity, transferability, composability, and linearity. These properties of LOCO Edit benefit greatly from the low-dimensional semantic subspace. Our method can further be extended to unsupervised or text-supervised editing in various text-to-image diffusion models (T-LOCO Edit). Finally, extensive empirical experiments demonstrate the effectiveness and efficiency of LOCO Edit. The code and the arXiv version can be found on the project website.[1]

## 1   Introduction

Recently, diffusion models have emerged as a powerful new family of deep generative models with remarkable performance in many applications such as image generation across various domains [1, 2, 3, 4, 5, 6], audio synthesis [7, 8], solving inverse problem [9, 10, 11, 12, 13, 14], and video generation [15, 16, 17]. For example, recent advances in AI-based image generation, revolutionized by diffusion models such as Dalle-2 [18], Imagen [19], and stable diffusion [4], have taken the world of "AI Art generation", enabling the generation of images directly from descriptive text inputs. These models corrupt images by adding noise through multiple steps of forward process and then generate samples by progressive denoising through multiple steps of the reverse generative process.

Although modern diffusion models are capable of generating photorealistic images from text prompts, manipulating the generated content by diffusion models in practice has remaining challenges. Unlike generative adversarial networks [20], the understanding of semantic spaces in diffusion models is still limited. Thus, achieving disentangled and localized control over content generation by direct manipulation of the semantic spaces remains a difficult task for diffusion models. Although effective, some existing editing methods in diffusion models often demand additional training procedures and are limited to global control of content generation [21, 22, 23]. Some methods are training-free or localized but are still based upon heuristics, lacking clear mathematical interpretations, or for text-supervised editing only [24, 25, 26, 27, 28]. Others provide analysis in diffusion models [29, 30, 31, 32, 33], but also have difficulty in local edits such as hair color.

---

[1]https://chicychen.github.io/LOCO

38th Conference on Neural Information Processing Systems (NeurIPS 2024).

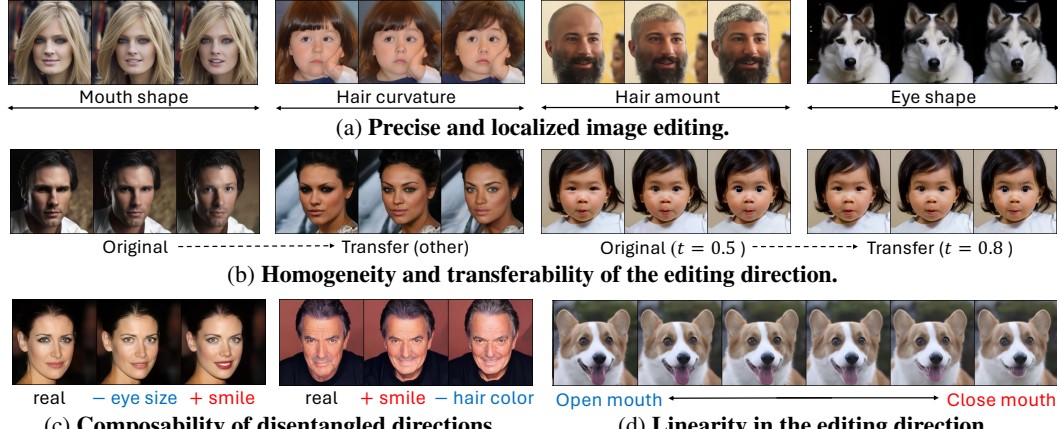

Figure 1: **LOCO Edit.** (a) The proposed method can perform precise localized editing in the region of interest. The editing direction is (b) homogeneous, (c) composable, and (d) linear.

In this study, we address the above problem by studying the low-rank semantic subspaces in diffusion models and proposing the LOw-rank COntrollable edit (LOCO Edit) approach. LOCO is the first local editing method that is single-step, training-free, requiring no text supervision, and having other intriguing properties (see Figure 1 for an illustration). Our method is highly intuitive and theoretically grounded, originating from a simple while intriguing observation in the learned *posterior mean predictor* (PMP) in diffusion models: for a large portion of denoising time steps,

> ***The PMP is a locally linear mapping between the noise image and the estimated clean image, and the singular vectors of its Jacobian reside within low-dimensional subspaces.***

The empirical evidence in Figure 2 consistently shows that this phenomenon occurs when training diffusion models using different network architectures on a range of real-world image datasets. Theoretically, we validated this observation by assuming a mixture of low-rank Gaussian distributions for the data. We then prove the local linearity of the PMP, the low-rank nature of its Jacobian, and that the singular vectors of the Jacobian span the low-dimensional subspaces.

By utilizing the linearity of the PMP, we can edit within the singular vector subspace of its Jacobian to achieve linear control of the image content with no label or text supervision. The editing direction can be efficiently computed using the generalized power method (GPM) [30, 34]. Furthermore, we can manipulate specific regions of interest in the image along a disentangled direction through efficient nullspace projection, taking advantage of the low-rank properties of the Jacobian.

**Benefits of LOCO Edit.** Compared to existing editing methods (e.g., [29, 35, 23, 24]) based on diffusion models, the proposed LOCO Edit offers several benefits that we highlight below:

- **Precise, single-step, training-free, and unsupervised editing.** LOCO enables precise *localized* editing (Figure 1a) in a single timestep without any training. Further, it requires no text supervision based on CLIP [36], thus integrating no intrinsic biases or flaws from CLIP [37]. LOCO is applicable to various diffusion models and datasets (Figure 5).

- **Linear, transferable, and composable editing directions.** The identified editing direction is linear, meaning that changes along this direction produce proportional changes in a semantic feature in the image space (Figure 1d). These editing directions are homogeneous and can be transferred across various images and noise levels (Figure 1b). Moreover, combining disentangled editing directions leads to simultaneous semantic changes in the respective region, while maintaining consistency in other areas (Figure 1c).

- **An intuitive and theoretically grounded approach.** Unlike previous works, by leveraging the local linearity of the PMP and the low-rankness of its Jacobian, our method is highly interpretable. The identified properties are well supported by both our empirical observation (Figure 2) and theoretical justifications in Section 4.

Moreover, LOCO Edit is generalizable to T-LOCO Edit for T2I diffusion models including DeepFloyd IF [19], Stable Diffusion [4], and Latent Consistency Models [38], with or without text supervision (Figure 4). A more detailed discussion on the relationship with prior arts can be found in Appendix B.

**Notations.** Throughout the paper, we use $\mathcal{X}_t \subseteq \mathbb{R}^d$ to denote the noise-corrupted image space at the time-step $t \in [0, 1]$. In particular, $\mathcal{X}_0$ denotes the clean image space with distribution $p_{\text{data}}(\boldsymbol{x})$, and $\boldsymbol{x}_0 \in \mathcal{X}_0$ denote an image. $\mathcal{X}_{0,t}$ denote the posterior mean space at time-step $t \in (0, 1]$. Here, $\mathbb{S}^{d-1}$ denotes a unit hypersphere in $\mathbb{R}^d$, and $\text{St}(d, r) := \{\boldsymbol{Z} \in \mathbb{R}^{d \times r} \mid \boldsymbol{Z}^\top \boldsymbol{Z} = \boldsymbol{I}_r\}$ denotes the Stiefel manifold. $\widetilde{\text{rank}}(\boldsymbol{A})$ denotes the numerical rank of $\boldsymbol{A}$. $\mathbb{E}_{\boldsymbol{x}_0 \sim p_{\text{data}}(\boldsymbol{x})}[\boldsymbol{x}_0|\boldsymbol{x}_t]$ denotes the posterior mean and is written as $\mathbb{E}[\boldsymbol{x}_0|\boldsymbol{x}_t]$. range(A) denotes the span of the columns of $\boldsymbol{A}$. null($\boldsymbol{A}$) denotes the set of solutions to $\boldsymbol{A}\boldsymbol{x} = 0$. $\text{proj}_{\text{null } \boldsymbol{A}}(\boldsymbol{x})$ denotes the projection of $\boldsymbol{x}$ onto null($\boldsymbol{A}$).

## 2 Preliminaries on Diffusion Models

In this section, we start by reviewing the basics of diffusion models [1, 2, 39], followed by several key techniques that will be used in our approach, such as Denoising Diffusion Implicit Models (DDIM) [3] and its inversion [40], T2I diffusion model, and classifier-free guidance [41].

**Basics of Diffusion Models.** In general, diffusion models consist of two processes:

- *The forward diffusion process.* The forward process progressively perturbs the original data $\boldsymbol{x}_0$ to a noisy sample $\boldsymbol{x}_t$ for $t \in [0, 1]$ with the Gaussian noise. As in [1], this can be characterized by a conditional Gaussian distribution $p_t(\boldsymbol{x}_t|\boldsymbol{x}_0) = \mathcal{N}(\boldsymbol{x}_t; \sqrt{\alpha_t}\boldsymbol{x}_0, (1-\alpha_t)\boldsymbol{I}_d)$. Particularly, parameters $\{\alpha_t\}_{t=0}^1$ sastify: (i) $\alpha_0 = 1$, and thus $p_0 = p_{\text{data}}$, and (ii) $\alpha_1 = 0$, and thus $p_1 = \mathcal{N}(\boldsymbol{0}, \boldsymbol{I}_d)$.

- *The reverse sampling process.* To generate a new sample, previous works [1, 3, 42, 43] have proposed various methods to approximate the reverse process of diffusion models. Typically, these methods involve estimating the noise $\boldsymbol{\epsilon}_t$ and removing the estimated noise from $\boldsymbol{x}_t$ recursively to obtain an estimate of $\boldsymbol{x}_0$. Specifically, the sampling step from $\boldsymbol{x}_t$ to $\boldsymbol{x}_{t-\Delta t}$ with a small $\Delta t > 0$ can be described as:

$$\boldsymbol{x}_{t-\Delta t} = \sqrt{\alpha_{t-\Delta t}} \left( \frac{\boldsymbol{x}_t - \sqrt{1-\alpha_t}\boldsymbol{\epsilon}_{\boldsymbol{\theta}}(\boldsymbol{x}_t, t)}{\sqrt{\alpha_t}} \right) + \sqrt{1-\alpha_{t-\Delta t}}\boldsymbol{\epsilon}_{\boldsymbol{\theta}}(\boldsymbol{x}_t, t), \tag{1}$$

where $\boldsymbol{\epsilon}_{\boldsymbol{\theta}}(\boldsymbol{x}_t, t)$ is parameterized by a neural network and trained to predict the noise at time $t$.

**Denoiser and Posterior Mean Predictor (PMP).** According to [1], the denoiser $\boldsymbol{\epsilon}_{\boldsymbol{\theta}}(\boldsymbol{x}_t, t)$ is optimized by solving the following problem:

$$\min_{\boldsymbol{\theta}} \ell(\boldsymbol{\theta}) := \mathbb{E}_{t \sim [0,1], \boldsymbol{x}_t \sim p_t(\boldsymbol{x}_t|\boldsymbol{x}_0), \boldsymbol{\epsilon} \sim \mathcal{N}(\boldsymbol{0},\boldsymbol{I})} \left[ \|\boldsymbol{\epsilon}_{\boldsymbol{\theta}}(\boldsymbol{x}_t, t) - \boldsymbol{\epsilon}\|_2^2 \right],$$

where $\boldsymbol{\theta}$ denotes the network parameters of the denoiser. Once $\boldsymbol{\epsilon}_{\boldsymbol{\theta}}$ is well trained, recent studies [44, 45] show that the posterior mean $\mathbb{E}[\boldsymbol{x}_0|\boldsymbol{x}_t]$, i.e., predicted clean image at time $t$, can be estimated as follows:

$$\hat{\boldsymbol{x}}_{0,t} = \boldsymbol{f}_{\boldsymbol{\theta},t}(\boldsymbol{x}_t; t) := \frac{\boldsymbol{x}_t - \sqrt{1-\alpha_t}\boldsymbol{\epsilon}_{\boldsymbol{\theta}}(\boldsymbol{x}_t, t)}{\sqrt{\alpha_t}}, \tag{2}$$

Here, $\boldsymbol{f}_{\boldsymbol{\theta},t}(\boldsymbol{x}_t; t)$ denotes the *posterior mean predictor* (PMP) [45, 44], and $\hat{\boldsymbol{x}}_{0,t} \in \mathcal{X}_{0,t}$ denotes the estimated posterior mean output from PMP given $\boldsymbol{x}_t$ and $t$ as the input. For simplicity, we denote $\boldsymbol{f}_{\boldsymbol{\theta},t}(\boldsymbol{x}_t; t)$ as $\boldsymbol{f}_{\boldsymbol{\theta},t}(\boldsymbol{x}_t)$.

**DDIM and DDIM Inversion.** Given a noisy sample $\boldsymbol{x}_t$ at time $t$, DDIM [3] can generate clean images by multiple denoising steps. Given a clean sample $\boldsymbol{x}_0$, DDIM inversion [3] can generate a noisy $\boldsymbol{x}_t$ at time $t$ by adding multiple steps of noise following the reversed trajectory of DDIM. DDIM inversion has been widely in image editing methods [40, 46, 29, 35, 47, 26] to obtain $\boldsymbol{x}_t$ given the original $\boldsymbol{x}_0$ and then performing editing starting from $\boldsymbol{x}_t$. In our work, after getting $\boldsymbol{x}_t$ given $\boldsymbol{x}_0$ via DDIM inversion, we edit $\boldsymbol{x}_t$ to $\boldsymbol{x}_t'$ only at the single time step $t$ with the help of PMP, and then utilize DDIM to generate the edited image $\boldsymbol{x}_0'$.

For ease of exposition, for any $t_1$ and $t_2$ with $t_2 > t_1$, we denote DDIM operator and its inversion as
$$\boldsymbol{x}_{t_1} = \text{DDIM}(\boldsymbol{x}_{t_2}, t_1) \quad \text{and} \quad \boldsymbol{x}_{t_2} = \text{DDIM-Inv}(\boldsymbol{x}_{t_1}, t_2).$$

**Text-to-image (T2I) Diffusion Models & Classifier-Free Guidance.** So far, our discussion has only focused on unconditional diffusion models. Moreover, our approach can be generalized from unconditional diffusion models to T2I diffusion models [38, 4, 48, 19], where the latter enables controllable image generation $\boldsymbol{x}_0$ guided by a text prompt $c$. In more detail, when training T2I diffusion models, we optimize a conditional denoising function $\boldsymbol{\epsilon}_{\boldsymbol{\theta}}(\boldsymbol{x}_t, t, c)$. For sampling, we

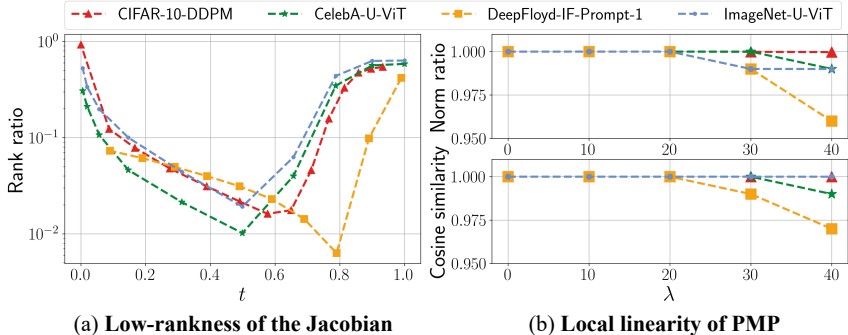

(a) **Low-rankness of the Jacobian**          (b) **Local linearity of PMP**

Figure 2: **Low-rankness of the Jacobian $J_{\theta,t}(x_t)$ and Local linearity of the PMP $f_{\theta,t}(x_t)$.** We evaluated DDPM (U-Net [49]) on CIFAR-10 dataset [50], U-ViT [51] (Transformer) on CelebA [52], ImageNet [53] datasets and DeepFloy IF [19] trained on LAION-5B [54] dataset. (a) The rank ratio of $J_{\theta,t}(x_t)$ against timestep $t$. (b) The norm ratio (Top) and cosine similarity (Bottom) between $f_{\theta,t}(x_t + \lambda\Delta x)$ and $l_{\theta}(x_t; \lambda\Delta x)$ against step size $\lambda$ at timestep $t = 0.7$.

employ a technique called *classifier-free guidance* [41], which substitutes the unconditional denoiser $\epsilon_{\theta}(x_t, t)$ in Equation (1) with its conditional counterpart $\tilde{\epsilon}_{\theta}(x_t, t, c)$ that can be described as follows:

$$\tilde{\epsilon}_{\theta}(x_t, t, c) = \epsilon_{\theta}(x_t, t, \varnothing) + \eta(\epsilon_{\theta}(x_t, t, c) - \epsilon_{\theta}(x_t, t, \varnothing)). \tag{3}$$

Here, $\varnothing$ denotes the empty prompt and $\eta > 0$ denotes the strength for the classifier-free guidance.

## 3  Exploring Linearity & Low-Dimensionality for Image Editting

In this section, we formally introduce the identified low-rank subspace in diffusion models and the proposed LOCO Edit method with the underlying intuitions. In Section 3.1, we present the benign properties in PMP that our method utilizes. Followed by this, in Section 3.3 we provide a detailed description of our method.

### 3.1  Local Linearity and Intrinsic Low-Dimensionality in PMP

First, let us delve into the key intuitions behind the proposed LOCO Edit method, which lie in the benign properties of the PMP $f_{\theta,t}(x_t)$. At one given timestep $t \in [0, 1]$, let us consider the first-order Taylor expansion of $f_{\theta,t}(x_t + \lambda\Delta x)$ at the point $x_t$:

$$\boxed{l_{\theta}(x_t; \lambda\Delta x) := f_{\theta,t}(x_t) + \lambda J_{\theta,t}(x_t) \cdot \Delta x,} \tag{4}$$

where $\Delta x \in \mathbb{S}^{d-1}$ is a perturbation direction with unit length, $\lambda \in \mathbb{R}$ is the perturbation strength, and $J_{\theta,t}(x_t) = \nabla_{x_t} f_{\theta,t}(x_t)$ is the Jacobian of $f_{\theta,t}(x_t)$. Interestingly, we discovered that within a certain range of noise levels, the learned PMP $f_{\theta,t}$ exhibits local linearity, and the singular subspace of its Jacobian $J_{\theta,t}$ is low rank. Notably, these properties are universal across various network architectures (e.g., UNet and Transformers) and datasets.

We measure the low-rankness with rank ratio and the local linearity with norm ratio and cosine similarity. Specifically, (*i*) *rank ratio* is the ratio of $\widetilde{\mathrm{rank}}(J_{\theta,t}(x_t))$ and the ambient dimension $d$; (*ii*) *norm ratio* is the ratio of $\|f_{\theta,t}(x_t + \lambda\Delta x)\|_2$ and $\|l_{\theta}(x_t; \lambda\Delta x)\|_2$; (*iii*) *cosine similarity* is between $f_{\theta,t}(x_t + \lambda\Delta x)$ and $l_{\theta}(x_t; \lambda\Delta x)$. The detailed experiment settings are provided in Appendix D.1, and results are illustrated in Figure 2, from which we observe:

- **Low-rankness of the Jacobian $J_{\theta,t}(x_t)$.** As shown in Figure 2(a), the *rank ratio* for $t \in [0, 1]$ *consistently* displays a U-shaped pattern across various network architectures and datasets: (*i*) it is close to 1 near either the pure noise $t = 1$ or the clean image $t = 0$, (*ii*) $J_{\theta,t}(x_t)$ is low-rank (i.e., rank ratio less than $10^{-1}$) for all diffusion models within the range $t \in [0.2, 0.7]$, (*iii*) it achieves the lowest value around mid-to-late timestep, slightly differs depending on architecture and dataset.

- **Local linearity of the PMP $f_{\theta,t}(x_t)$.** Moreover, the mapping $f_{\theta,t}(x_t)$ exhibits strong linearity across a large portion of the timesteps; see Figure 2(b) and Figure 10. Specifically, in Figure 2(b), we evaluate the linearity of $f_{\theta,t}(x_t)$ at $t = 0.7$ where the rank ratio is close to the lowest value. We can see that $f_{\theta,t}(x_t + \lambda\Delta x) \approx l_{\theta}(x_t; \lambda\Delta x)$ even when $\lambda = 40$, which is consistently true among different architectures trained on different datasets.

In addition to comprehensive experimental studies, we will also demonstrate in Section 4 that both properties can be theoretically justified.

## 3.2 Key Intuitions for Our Image Editing Method

The two benign properties offer valuable insights for image editing with precise control. Here, we first present the high-level intuitions behind our method, with further details postponed to Section 3.3. Specifically, for any given time-step $t \in [0, 1]$, let us denote the compact singular value decomposition (SVD) of the Jacobian $\boldsymbol{J}_{\boldsymbol{\theta},t}(\boldsymbol{x}_t)$ as

$$\boldsymbol{J}_{\boldsymbol{\theta},t}(\boldsymbol{x}_t) \;=\; \boldsymbol{U}\boldsymbol{\Sigma}\boldsymbol{V}^\top \;=\; \sum_{i=1}^r \sigma_i \boldsymbol{u}_i \boldsymbol{v}_i^\top, \tag{5}$$

where $r$ is the rank of $\boldsymbol{J}_{\boldsymbol{\theta},t}(\boldsymbol{x}_t)$, $\boldsymbol{U} = [\boldsymbol{u}_1 \;\; \cdots \;\; \boldsymbol{u}_r] \in \mathrm{St}(d,r)$ and $\boldsymbol{V} = [\boldsymbol{v}_1 \;\; \cdots \;\; \boldsymbol{v}_r] \in \mathrm{St}(d,r)$ denote the left and right singular vectors, and $\boldsymbol{\Sigma} = \mathrm{diag}(\sigma_1, \cdots, \sigma_r)$ denote the singular values. We write $\boldsymbol{J}_{\boldsymbol{\theta},t}(\boldsymbol{x}_t) = \boldsymbol{J}_{\boldsymbol{\theta},t}$ in short for a specific $\boldsymbol{x}_t$, and denote $\mathrm{range}(\boldsymbol{J}_{\boldsymbol{\theta},t}^\top) = \mathrm{span}(\boldsymbol{V})$ and $\mathrm{null}(\boldsymbol{J}_{\boldsymbol{\theta},t}) = \{\boldsymbol{w} \mid \boldsymbol{J}_{\boldsymbol{\theta},t}\boldsymbol{w} = 0\}$.

- **Local linearity of PMP for one-step, training-free, and supervision-free editing.** Given the PMP $\boldsymbol{f}_{\boldsymbol{\theta},t}(\boldsymbol{x}_t)$ is locally linear at the $t$-th timestep, if we perturb $\boldsymbol{x}_t$ by $\Delta\boldsymbol{x} = \lambda\boldsymbol{v}_i$, using one right singular vector $\boldsymbol{v}_i$ of $\boldsymbol{J}_{\boldsymbol{\theta},t}(\boldsymbol{x}_t)$ as an example editing direction, then by orthogonality

$$\boldsymbol{f}_{\boldsymbol{\theta},t}(\boldsymbol{x}_t + \lambda\boldsymbol{v}_i) \;\approx\; \boldsymbol{f}_{\boldsymbol{\theta},t}(\boldsymbol{x}_t) + \boldsymbol{J}_{\boldsymbol{\theta},t}(\boldsymbol{x}_t)\boldsymbol{v}_i \;=\; \boldsymbol{f}_{\boldsymbol{\theta},t}(\boldsymbol{x}_t) + \lambda\sigma_i\boldsymbol{u}_i \;=\; \hat{\boldsymbol{x}}_{0,t} + \rho_i\boldsymbol{u}_i. \tag{6}$$

  This implies we can achieve *one-step editing* along the semantic direction $\boldsymbol{u}_i$. Notably, the method is training-free and supervision-free since $\boldsymbol{v}_i$ can be simply found via the SVD of $\boldsymbol{J}_{\boldsymbol{\theta},t}(\boldsymbol{x}_t)$.

- **Local linearity of PMP for linear, homogeneous, and composable image editing.** (*i*) First, the editing direction $\boldsymbol{v} = \boldsymbol{v}_i$ is *linear*, where any linear $\lambda \in \mathbb{R}$ change along $\boldsymbol{v}_i$ results in a linear change $\rho_i = \lambda\sigma_i$ along $\boldsymbol{u}_i$ for the edited image. (*ii*) Second, the editing direction $\boldsymbol{v} = \boldsymbol{v}_i$ is *homogeneous* due to its independence of $\hat{\boldsymbol{x}}_{0,t}$, where it could be applied on any images from the same data distribution and results in the same semantic editing. (*iii*) Third, editing directions are *composable*. Any linearly combined editing direction $\boldsymbol{v} = \sum_{i\in\mathcal{I}} \lambda_i\boldsymbol{v}_i \in \mathrm{range}\left(\boldsymbol{J}_{\theta,t}^\top\right)$ is a valid editing direction which would result in a composable change $\sum_{i\in\mathcal{I}} \rho_i\boldsymbol{u}_i$ in the edited image. On the contrary, $\boldsymbol{w} \in \mathrm{null}\left(\boldsymbol{J}_{\boldsymbol{\theta},t}\right)$ results in no editing since $\boldsymbol{f}_{\boldsymbol{\theta},t}(\boldsymbol{x}_t + \lambda\boldsymbol{w}) \approx \boldsymbol{f}_{\boldsymbol{\theta},t}(\boldsymbol{x}_t)$.

- **Low-rankness of Jacobian for localized and efficient editing.** $\boldsymbol{J}_{\boldsymbol{\theta},t}(\boldsymbol{x}_t)$ is for the entire predicted clean image, thus $\boldsymbol{J}_{\boldsymbol{\theta},t}(\boldsymbol{x}_t)$ finds editing directions in the entire image. Denote $\tilde{\boldsymbol{J}}_{\boldsymbol{\theta},t}$ the Jacobian only for a certain region of interest (ROI), and $\bar{\boldsymbol{J}}_{\boldsymbol{\theta},t}$ the Jacobian for regions outside ROI. Similarly, $\boldsymbol{v} \in \mathrm{range}\left(\tilde{\boldsymbol{J}}_{\theta,t}^\top\right)$ can edit mainly regions within the ROI, and $\mathrm{null}\left(\bar{\boldsymbol{J}}_{\theta,t}^\top\right)$ contain directions that do not edit regions outside of ROI. Further projection of $\boldsymbol{v}$ onto $\mathrm{null}\left(\bar{\boldsymbol{J}}_{\theta,t}^\top\right)$ can result in a more localized editing direction for ROI. To perform such nullspace projection, computing the full SVD can be very expensive. But we can highly reduce the computation by the low-rank estimation of Jacobians with rank $r' \ll d$. The estimation is efficient yet effective with $t \in [0.5, 0.7]$ when the rank of the Jacobian achieves the lowest value.

## 3.3 Low-rank Controllable Image Editing Method with Nullspace Projection

In this subsection, we provide a detailed introduction to LOCO Edit, expanding on the discussion in Section 3.1. We first introduce the supervision-free LOCO Edit, where we further enable localized image editing through nullspace projection with masks. Second, we generalize to T-LOCO Edit for T2I diffusion models w/wo text-supervision to define the semantic editing directions.

**LOCO Edit.**  We first introduce the general pipeline of LOCO Edit. As illustrated in Figure 3, given an original image $\boldsymbol{x}_0$, we first use $\boldsymbol{x}_t = \texttt{DDIM-Inv}(\boldsymbol{x}_0, t)$ to generate a noisy image $\boldsymbol{x}_t$. In particular, we choose $t \in [0.5, 0.7]$ so that the PMP $\boldsymbol{f}_{\boldsymbol{\theta},t}(\boldsymbol{x}_t)$ is locally linear and its Jacobian $\boldsymbol{J}_{\boldsymbol{\theta},t}(\boldsymbol{x}_t)$ is close to its lowest rank. From Section 3.1, we know that we can edit the image by changing $\boldsymbol{x}_t' = \boldsymbol{x}_t + \lambda\boldsymbol{v}_p$, where $\boldsymbol{v}_p$ is the identified editing direction. After editing $\boldsymbol{x}_t$ to $\boldsymbol{x}_t'$, we use $\boldsymbol{x}_0' = \texttt{DDIM}\left(\boldsymbol{x}_t', 0\right)$ to generate the edited image.
In many practical applications, we often need to edit only specific *local* regions of an image while leaving the rest unchanged. As discussed in Section 3.2, we can achieve this task by finding a precise local editing direction with localized Jacobians and nullspace projection. Overall, the complete method is in Algorithm 1. We describe the key details as follows.

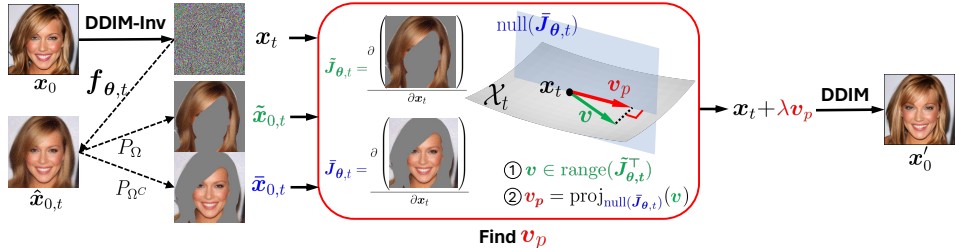

Figure 3: **Illustration of the unsupervised LOCO Edit for unconditional diffusion models.** Given an image $\boldsymbol{x}_0$, we perform DDIM-Inv until time $t$ to get $\boldsymbol{x}_t$, and estimate $\hat{\boldsymbol{x}}_{0,t}$ from $\boldsymbol{x}_t$. After masking to get the region of interest (ROI) $\tilde{\boldsymbol{x}}_{0,t}$ and its counterparts $\bar{\boldsymbol{x}}_{0,t}$, we find the edit direction $\boldsymbol{v}_p$ via SVD and nullspace projection based on their Jacobians (Algorithm 1). By denoising $\boldsymbol{x}_t + \lambda \boldsymbol{v}_p$, an image $\boldsymbol{x}_0'$ with localized editing is generated. *In this paper, the variables and notions related to ROI, nullspace, and final direction are respectively highlighted by green, blue, and red colors.*

---

**Algorithm 1** Unsupervised LOCO Edit

1: **Input**: original image $\boldsymbol{x}_0$, the mask $\Omega$, pretrained diffusion model $\boldsymbol{\epsilon}_{\boldsymbol{\theta}}$, editing strength $\lambda$, semantic index $k$, number of semantic directions $r$, editing timestep $t \in [0.5, 0.7]$, the rank $r' = 5$.

2: **Output**: edited image $\boldsymbol{x}_0'$,

3: Generate $\boldsymbol{x}_t \leftarrow \texttt{DDIM-Inv}(\boldsymbol{x}_0, t)$              $\triangleright$ noisy image at $t$-th timestep

4: Compute the top-$r$ SVD $(\tilde{\boldsymbol{U}}, \tilde{\boldsymbol{\Sigma}}, \tilde{\boldsymbol{V}})$ of $\tilde{\boldsymbol{J}}_{\boldsymbol{\theta},t} = \nabla_{\boldsymbol{x}_t} P_\Omega(\boldsymbol{f}_{\boldsymbol{\theta},t}(\boldsymbol{x}_t))$

5: Compute the top-$r'$ SVD $(\bar{\boldsymbol{U}}, \bar{\boldsymbol{\Sigma}}, \bar{\boldsymbol{V}})$ of $\bar{\boldsymbol{J}}_{\boldsymbol{\theta},t} = \nabla_{\boldsymbol{x}_t} P_{\Omega^C}(\boldsymbol{f}_{\boldsymbol{\theta},t}(\boldsymbol{x}_t))$

6: Pick direction $\boldsymbol{v} \leftarrow \tilde{\boldsymbol{V}}[:, i]$      $\triangleright$ ① Pick the $k^{th}$ singular vector for the editing direction

7: Compute $\boldsymbol{v}_p \leftarrow (\boldsymbol{I} - \bar{\boldsymbol{V}} \bar{\boldsymbol{V}}^\top) \cdot \boldsymbol{v}$      $\triangleright$ ② Nullspace projection for editing within the mask $\Omega$

8: $\boldsymbol{v}_p \leftarrow \boldsymbol{v}_p / \|\boldsymbol{v}_p\|_2$            $\triangleright$ Normalize the editing direction

9: **Return**: $\boldsymbol{x}_0' \leftarrow \texttt{DDIM}(\boldsymbol{x}_t + \lambda \boldsymbol{v}_p, 0)$      $\triangleright$ Editing with forward DDIM along the direction $\boldsymbol{v}_p$

---

- **Finding localized Jacobians via masking.** To enable local editing, we use a mask $\Omega$ (i.e., an index set of pixels) to select the region of interest,[2] with $\mathcal{P}_\Omega(\cdot)$ denoting the projection onto the index set $\Omega$. For picking a local editing direction, we calculate the Jacobian of $\boldsymbol{f}_{\boldsymbol{\theta},t}(\boldsymbol{x}_t)$ restricted to the region of interest, $\tilde{\boldsymbol{J}}_{\boldsymbol{\theta},t} = \nabla_{\boldsymbol{x}_t} P_\Omega(\boldsymbol{f}_{\boldsymbol{\theta},t}(\boldsymbol{x}_t)) = \tilde{\boldsymbol{U}} \tilde{\boldsymbol{\Sigma}} \tilde{\boldsymbol{V}}^\top$, and select the localized editing direction $\boldsymbol{v}$ from the top-$r$ singular vectors of $\tilde{\boldsymbol{V}}$ (e.g., $\boldsymbol{v} = \tilde{\boldsymbol{V}}[:, k] \in \text{range} \tilde{\boldsymbol{J}}_{\boldsymbol{\theta},t}^\top$ for some index $k \in [r]$). In practice, a top-$r$ rank estimation for $\tilde{\boldsymbol{V}}$ is calculated through the generalized power method (GPM) Algorithm 2 with $r = 5$ to improve efficiency.

- **Better semantic disentanglement via nullspace projection.** However, the projection $\mathcal{P}_\Omega(\cdot)$ introduces extra *nonlinearity* into the mapping $P_\Omega(\boldsymbol{f}_{\boldsymbol{\theta},t}(\boldsymbol{x}_t))$, causing the identified direction to have semantic entanglements with the area $\Omega^C$ outside of the mask. Here, $\Omega^C$ denotes the complimentary set of $\Omega$. To address this issue, we can use the nullspace projection method [56, 57]. Specifically, given $\bar{\boldsymbol{J}}_{\boldsymbol{\theta},t} = \nabla_{\boldsymbol{x}_t} P_{\Omega^C}(\boldsymbol{f}_{\boldsymbol{\theta},t}(\boldsymbol{x}_t)) = \bar{\boldsymbol{U}} \bar{\boldsymbol{\Sigma}} \bar{\boldsymbol{V}}^\top$, nullspace projection projects $\boldsymbol{v}$ onto $\text{null}\left(\bar{\boldsymbol{J}}_{\theta,t}^\top\right)$. The projection can be computed as $\boldsymbol{v}_p = \text{proj}_{\text{null}\left(\bar{\boldsymbol{J}}_{\theta,t}^\top\right)}(\boldsymbol{v}) = (\boldsymbol{I} - \bar{\boldsymbol{V}} \bar{\boldsymbol{V}}^\top) \boldsymbol{v}$ so that the modified $\boldsymbol{v}_p$ does not change the image in $\Omega^C$. In practice, we calculate a top-$r'$ rank estimation for $\bar{\boldsymbol{V}}$ through the generalized power method (GPM) Algorithm 2 with $r' = 5$.

**T-LOCO Edit.** The unsupervised edit method can be seamlessly applied to T2I diffusion models with classifier-free guidance (3) (Algorithm 3). Besides, we can further enable text-supervised image editing with an editing prompt (Algorithm 4). See results in Figure 4(a). This is useful because the additional text prompt allows us to enforce a specified editing direction that *cannot* be found easily in the semantic subspace of the vanilla Jacobian $\boldsymbol{J}_{\boldsymbol{\theta},t}$. As illustrated in Figure 4(b), this includes adding glasses or changing the curly hair of a human face. For simplicity, we introduce the key ideas of text-supervised T-LOCO Edit based upon DeepFloyd IF [19]. Similar procedures are also generalized to Stable Diffusion and Latent Consistency Models with an additional decoding step [4, 38]. We discuss the key intuition below, see Appendix E.2 and Appendix E.3 for method details.

---

[2]For datasets that have predefined masks, we can use them directly. For other datasets that lack predefined masks as well as generate images, we can utilize Segment Anything (SAM) to generate masks [55].

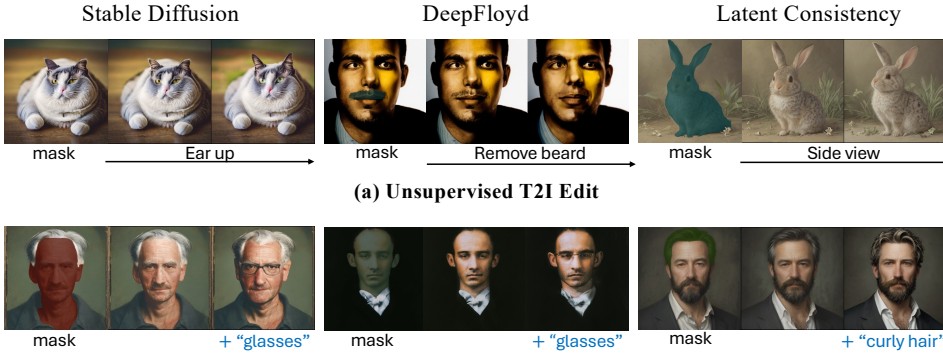

| Stable Diffusion | DeepFloyd | Latent Consistency |

mask — Ear up →  mask — Remove beard →  mask — Side view →

**(a) Unsupervised T2I Edit**

mask — + "glasses"  mask — + "glasses"  mask — + "curly hair"

**(b) Text-supervised T2I Edit**

Figure 4: **T-LOCO Edit on T2I diffusion models.** (a) Unsupervised editing direction is found only via the given mask without editing prompt. (b) Text-supervised editing direction is found with both a mask and an editing prompt such as "with glasses". Experiment details can be found in Appendix G.3.

We first introduce some notations. Let $c_o$ denote the original prompt, and $c_e$ denote the editing prompt. For example, in Figure 4(b), $c_o$ can be "portrait of a man", while $c_e$ can be "portrait of a man with glasses". Correspondingly, given the noisy image $\boldsymbol{x}_t$ for the clean image $\boldsymbol{x}_0$ generated with $c_o$, let $\boldsymbol{f}_{\boldsymbol{\theta},t}^o(\boldsymbol{x}_t)$ and $\boldsymbol{J}_{\boldsymbol{\theta},t}^o(\boldsymbol{x}_t)$ be the estimated posterior mean and its Jacobian conditioned on the original prompt $c_o$, and let $\boldsymbol{f}_{\boldsymbol{\theta},t}^e(\boldsymbol{x}_t)$ and $\boldsymbol{J}_{\boldsymbol{\theta},t}^e(\boldsymbol{x}_t)$ be the estimated posterior mean and its Jacobian conditioned on both the editing prompt $c_e$ and $c_o$.

According to the classifier-free guidance (3), we can estimate the *difference* of estimated posterior means caused by the editing prompt as $\boldsymbol{d} = \boldsymbol{f}_{\boldsymbol{\theta},t}^e(\boldsymbol{x}_t) - \boldsymbol{f}_{\boldsymbol{\theta},t}^o(\boldsymbol{x}_t)$, and then set $\boldsymbol{v} = \boldsymbol{J}_{\boldsymbol{\theta},t}^e(\boldsymbol{x}_t)^\top \boldsymbol{d}$ as an initial estimator of the editing direction.[3] Based upon this, to enable localized editing, similar to the unsupervised case, we can apply masks $\Omega$ to select ROI in $\boldsymbol{d}$ and calculate localized Jacobian to get $\boldsymbol{v}$. After that, similarly, we can perform nullspace projection of $\boldsymbol{v}$ for better disentanglement to get the final editing direction $\boldsymbol{v}_p$.

## 4 Justification of Local Linearity, Low-rankness, & Semantic Direction

In this section, we provide theoretical justification for the benign properties in Section 3.1. First, we assume that the image distribution $p_{\text{data}}$ follows *mixture of low-rank Gaussians* defined as follows.

**Assumption 1.** *The data $\boldsymbol{x}_0 \in \mathbb{R}^d$ generated distribution $p_{data}$ lies on a union of $K$ subspaces. The basis of each subspace $\{\boldsymbol{M}_k \in \text{St}(d, r_k)\}_{k=1}^K$ are orthogonal to each other with $\boldsymbol{M}_i^\top \boldsymbol{M}_j = \boldsymbol{0}$ for all $1 \le i \ne j \le K$, and the subspace dimension $r_k$ is much smaller than the ambient dimension $d$. Moreover, for each $k \in [K]$, $\boldsymbol{x}_0$ follows degenerated Gaussian with $\mathbb{P}(\boldsymbol{x}_0 = \boldsymbol{M}_k \boldsymbol{a}_k) = 1/K, \boldsymbol{a}_k \sim \mathcal{N}(\boldsymbol{0}, \boldsymbol{I}_{r_k})$. Without loss of generality, suppose $\boldsymbol{x}_t$ is from the $h$-th class, that is $\boldsymbol{x}_t = \sqrt{\alpha_t}\boldsymbol{x}_0 + \sqrt{1 - \alpha_t}\boldsymbol{\epsilon}$ where $\boldsymbol{x}_0 \in \text{range}(\boldsymbol{M}_h)$, i.e. $\boldsymbol{x}_0 = \boldsymbol{M}_h \boldsymbol{a}_h$. Both $\|\boldsymbol{x}_0\|_2, \|\boldsymbol{\epsilon}\|_2$ is bounded.*

Our data assumption is motivated by the intrinsic low-dimensionality of real-world image dataset [58].Additionally, Wang et al. [59] demonstrated that images generated by an analytical score function derived from a mixture of Gaussians distribution exhibit conceptual similarities to those produced by practically trained diffusion models. Given that $\boldsymbol{f}_{\boldsymbol{\theta},t}(\boldsymbol{x}_t)$ is an estimator of the posterior mean $\mathbb{E}[\boldsymbol{x}_0|\boldsymbol{x}_t]$, we show that the posterior mean $\mathbb{E}[\boldsymbol{x}_0|\boldsymbol{x}_t]$ can analytically derived as follows.

**Lemma 1.** *Under Assumption 1, for $t \in (0, 1]$, the posterior mean is*

$$\mathbb{E}[\boldsymbol{x}_0|\boldsymbol{x}_t] = \sqrt{\alpha_t} \frac{\sum_{k=1}^K \exp\left(\frac{\alpha_t}{2(1 - \alpha_t)}\|\boldsymbol{M}_k^\top \boldsymbol{x}_t\|^2\right) \boldsymbol{M}_k \boldsymbol{M}_k^\top \boldsymbol{x}_t}{\sum_{k=1}^K \exp\left(\frac{\alpha_t}{2(1 - \alpha_t)}\|\boldsymbol{M}_k^\top \boldsymbol{x}_t\|^2\right)}. \tag{7}$$

Lemma 1 shows that the posterior mean $\mathbb{E}[\boldsymbol{x}_0|\boldsymbol{x}_t]$ could be viewed as a convex combination of $\boldsymbol{M}_k \boldsymbol{M}_k^\top \boldsymbol{x}_t$, i.e. $\boldsymbol{x}_t$ projected onto each subspace $\boldsymbol{M}_k$. This lemma leads to the following theorem:

---

[3]The idea is to identify the editing direction in the $\mathcal{X}_t$ space based on changes in the estimated posterior mean caused by the editing prompt. More details are provided in Appendix E.3.

**Theorem 1.** *Based upon Assumption 1, we can show the following three properties for the posterior mean* $\mathbb{E}[\boldsymbol{x}_0|\boldsymbol{x}_t]$:

- *The Jacobian of posterior mean satisfies* $\mathrm{rank}\left(\nabla_{\boldsymbol{x}_t}\mathbb{E}[\boldsymbol{x}_0|\boldsymbol{x}_t]\right) \leq r := \sum_{k=1}^{K} r_k$ *for all* $t \in (0,1]$.

- *The posterior mean* $\mathbb{E}[\boldsymbol{x}_0|\boldsymbol{x}_t]$ *has local linearity such that*

$$\left\| \mathbb{E}\left[\boldsymbol{x}_0|\boldsymbol{x}_t + \lambda\Delta\boldsymbol{x}\right] - \mathbb{E}\left[\boldsymbol{x}_0|\boldsymbol{x}_t\right] - \lambda\nabla_{\boldsymbol{x}_t}\mathbb{E}[\boldsymbol{x}_0|\boldsymbol{x}_t] \cdot \Delta\boldsymbol{x} \right\| = \lambda\frac{\alpha_t}{(1-\alpha_t)}\mathcal{O}(\lambda), \qquad (8)$$

   *where* $\Delta\boldsymbol{x} \in \mathbb{S}^{d-1}$ *and* $\lambda \in \mathbb{R}$ *is the step size.*

- $\nabla_{\boldsymbol{x}_t}\mathbb{E}[\boldsymbol{x}_0|\boldsymbol{x}_t]$ *is symmetric and the full SVD of* $\nabla_{\boldsymbol{x}_t}\mathbb{E}[\boldsymbol{x}_0|\boldsymbol{x}_t]$ *could be written as* $\nabla_{\boldsymbol{x}_t}\mathbb{E}[\boldsymbol{x}_0|\boldsymbol{x}_t] = \boldsymbol{U}_t\boldsymbol{\Sigma}_t\boldsymbol{V}_t^{\top}$, *where* $\boldsymbol{U}_t = [\boldsymbol{u}_{t,1}, \boldsymbol{u}_{t,2}, \ldots, \boldsymbol{u}_{t,d}] \in \mathrm{St}(d,d)$, $\boldsymbol{\Sigma}_t = \mathrm{diag}(\sigma_{t,1}, \ldots, \sigma_{t,r}, \ldots, 0)$ *with* $\sigma_{t,1} \geq \cdots \geq \sigma_{t,r} \geq 0$ *and* $\boldsymbol{V}_t = [\boldsymbol{v}_{t,1}, \boldsymbol{v}_{t,2}, \ldots, \boldsymbol{v}_{t,d}] \in \mathrm{St}(d,d)$. *Let* $\boldsymbol{U}_{t,1} := [\boldsymbol{u}_{t,1}, \boldsymbol{u}_{t,2}, \ldots, \boldsymbol{u}_{t,r}]$ *and* $\boldsymbol{M} := [\boldsymbol{M}_1, \boldsymbol{M}_2, \ldots, \boldsymbol{M}_K]$. *It holds that* $\lim_{t \to 1}\left\|\left(\boldsymbol{I}_d - \boldsymbol{U}_{t,1}\boldsymbol{U}_{t,1}^{\top}\right)\boldsymbol{M}\right\|_F = 0$.

The proof is deferred to Appendix F. Admittedly, there are gap between our theory and practice, such as the approximation error between $\boldsymbol{f}_{\boldsymbol{\theta},t}(\boldsymbol{x}_t)$ and $\mathbb{E}[\boldsymbol{x}_0|\boldsymbol{x}_t]$, assumptions about the data distribution, and the high rankness of $\boldsymbol{J}_{\boldsymbol{\theta},t}$ for $t < 0.2$ and $t > 0.9$ in Figure 2. Nonetheless, Theorem 1 largely supports our empirical observation in Section 3 that we discuss below:

- **Low-rankness of the Jacobian.** The first property in Theorem 1 demonstrates that the rank of $\nabla_{\boldsymbol{x}_t}\mathbb{E}[\boldsymbol{x}_0|\boldsymbol{x}_t]$ is always no greater than the intrinsic dimension of the data distribution. Given that the intrinsic dimension of the real data distribution is usually much lower than the ambient dimension [58], the rank of $\boldsymbol{J}_{\boldsymbol{\theta},t}$ on the real dataset should also be low. The results align with our empirical observations in Figure 2 when $t \in [0.2, 0.7]$.

- **Linearity of the posterior mean.** The second property in Theorem 1 shows that the linear approximation error is within the order of $\lambda\alpha_t/(1-\alpha_t) \cdot \mathcal{O}(\lambda)$. This implies that when $t$ approaches 1, $\alpha_t/(1-\alpha_t)$ becomes small, resulting in a small approximation error even for large $\lambda$. Empirically, Figure 2 shows that the linear approximation error of $\boldsymbol{f}_{\boldsymbol{\theta},t}(\boldsymbol{x}_t)$ is small when $t = 0.7$ and $\lambda = 40$, whereas Figure 10 shows a much larger error for $t = 0.0$ under the same $\lambda$. These observations align well with our theory.

- **Low-dimensional semantic subspace.** The third property in Theorem 1 shows that, when $t$ is close to 1, left singular vectors associated with the top-$r$ singular values form the basis of the image distribution. Since the editing direction consists of basis, the edited image remains within the image distribution. This explains why $\boldsymbol{u}_i$ found in Equation (6) is a semantic direction for image editing.

## 5  Experiments

In this section, we perform extensive experiments to demonstrate the effectiveness and efficiency of LOCO Edit. We first showcase LOCO Edit has strong *localized* editing ability across a variety of datasets in Section 5.1. Moreover, we conduct comprehensive comparisons with other methods to show the superiority of the LOCO Edit method in Section 5.2. Besides, we provide ablation studies on multiple components in our method in Appendix C.1, and analyze the editing directions in Appendix C.2, with extra experimental details postponed to Appendix G.

### 5.1  Demonstration on Localized Editing and Other Benign Properties

First, we demonstrate benign properties of LOCO Edit in Algorithm 1 on a variety of datasets, including LSUN-Church [60], Flower [61], AFHQ [62], CelebA-HQ [52], and FFHQ [63].

As shown in Figure 5 and Figure 1a, our method enables editing specific localized regions such as eye size/focus, hair curvature, length/amount, and architecture, while preserving the consistency of other regions. Besides the ability of precise local editing, Figure 1 demonstrates the benign properties of the identified editing directions and verify our analysis in Section 4:

- **Linearity.** As shown Figure 1(d), the semantic editing can be strengthened through larger editing scales and can be flipped by negating the scale.

- **Homogeneity and transferability.** As shown Figure 1(b), the discovered editing direction can be transferred across samples and timesteps in $\mathcal{X}_t$.

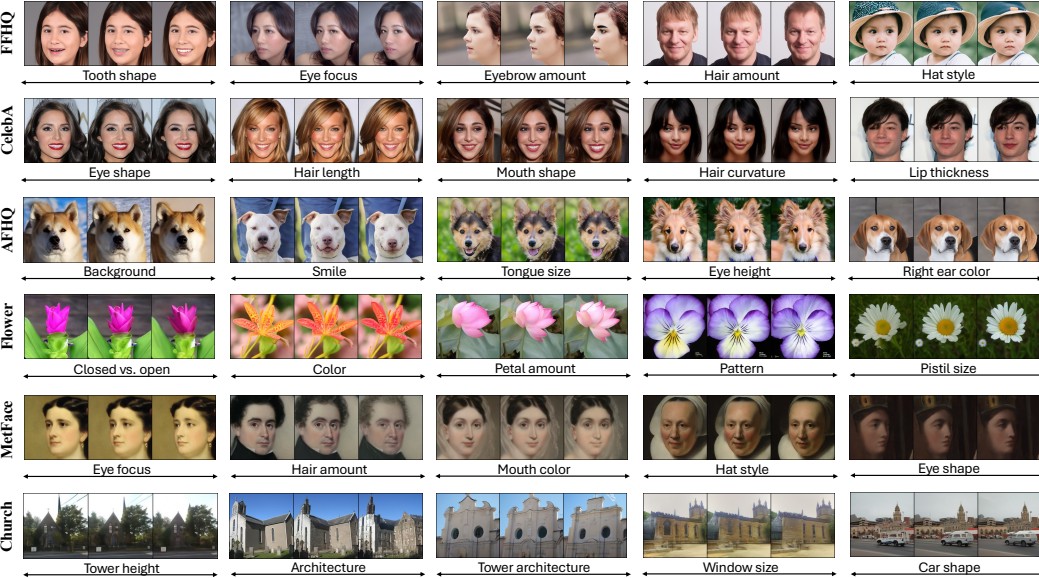

Figure 5: **Benchmarking LOCO Edit across various datasets.** For each group of three images, in the center is the original image, and on the left and right are edited images along the negative and the positive directions accordingly.

- **Composability.** As shown Figure 1(c), the identified disentangled editing directions in the low-rank subspace allow direct composition without influencing each other.

## 5.2 Comprehensive Comparison with Other Image Editing Methods

We compare LOCO Edit with several notable and recent image editing techniques, including Asyrp [29], Pullback [30], NoiseCLR [23], and BlendedDifusion [24]. We also compare with an unexplored method using the Jacobians $\frac{\partial \boldsymbol{\epsilon}_t}{\partial \boldsymbol{x}_t}$ to find the editing direction, named as $\frac{\partial \boldsymbol{\epsilon}_t}{\partial \boldsymbol{x}_t}$.

**Metrics.** We evaluate our method using the below metrics and summarize the results in Table 1. Besides the image generation quality, we also compared other attributes such as the local edit ability, efficiency, the requirement for supervision, and theoretical justifications.

- *Local Edit Success Rate* evaluates whether the editing successfully changes the target semantics and preserves unrelated regions by human evaluators.
- *LPIPS* [64] and *SSIM* [65] measure the consistency between edited and original images.
- *Transfer Success Rate* measures whether the editing transferred to other images successfully changes the target semantics and preserves unrelated regions by human evaluators.
- *Learning time* to measure the time required to identify the edit directions.
- *Transfer Edit Time* to measure the time required to transfer the editing to other images directly.
- *#Images for Learning* measures the number of images used to find the editing directions.
- *One-step Edit*, *No Additional Supervision*, *Theoretically Grounded*, and *Localized Edit* are attributes of the editing methods, where each of them measures a specific property for the method.

Moreover, we visualize the editing results on non-cherry-picked images in Figure 6. The detailed evaluation settings are provided in Appendix G.2.

**Benefits of Our Method.** Based upon the qualitative and quantitative comparisons, our method shows several clear advantages that we summarize as follows.

- **Superior local edit ability with one-step edit.** Table 1 shows LOCO Edit achieves the best Local Edit Success Rate. Such local edit ability only requires one-step edit at a specific time $t$. For LPIPS and SSIM, our method performs better than most methods but worse than BlendedDiffusion. However, BlendedDiffusion sometimes fails the edit within the masks (as visualized in Figure 6, rows 1, 3, 4, and 5). Other methods find semantic direction more globally, leading to worse performance in Local Edit Success Rate, LPIPS, and SSIM for localized edits.

| Method Name | Pullback | $\partial\boldsymbol{\epsilon}_t/\partial\boldsymbol{x}_t$ | NoiseCLR | Asyrp | BlendedDiffusion | **LOCO (Ours)** |
|---|---|---|---|---|---|---|
| Local Edit Success Rate↑ | 0.32 | 0.37 | 0.32 | 0.47 | **0.55** | **0.80** |
| LPIPS↓ | 0.16 | 0.13 | 0.14 | 0.22 | **0.03** | **0.08** |
| SSIM↑ | 0.60 | 0.66 | 0.68 | 0.68 | **0.94** | **0.71** |
| Transfer Success Rate↑ | 0.14 | 0.24 | **0.66** | 0.58 | Can't Transfer | **0.91** |
| Transfer Edit Time↓ | 4s | **2s** | 5s | 3s | Can't Transfer | **2s** |
| #Images for Learning | **1** | **1** | 100 | 100 | **1** | **1** |
| Learning Time↓ | **8s** | **44s** | 1 day | 475s | 120s | 79s |
| One-step Edit? | ✓ | ✓ | ✗ | ✗ | ✗ | ✓ |
| No Additional Supervision? | ✓ | ✓ | ✓ | ✗ | ✗ | ✓ |
| Theoretically Grounded? | ✗ | ✗ | ✗ | ✗ | ✗ | ✓ |
| Localized Edit? | ✗ | ✗ | ✗ | ✗ | ✓ | ✓ |

Table 1: **Comparisons with existing methods.** Our LOCO Edit excels in localized editing, transferability and efficiency, with other intriguing properties such as one-step edit, supervision-free, and theoretically grounded.

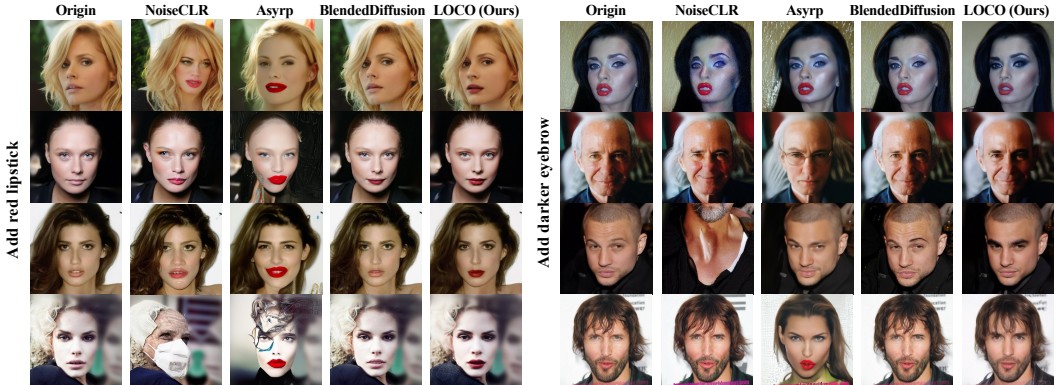

Figure 6: **Compare local edit ability with other works on non-cherry-picked images.** LOCO has consistent and accurate local edit ability, while other methods have wrong, global, or no edits.

- **Transferability and efficiency.** First, LOCO Edit requires less learning time than most of the other methods and requires learning only for a single time step with a single image. Moreover, LOCO Edit is highly transferable, having the highest Transfer Success Rate in Table A. In contrast, BlendedDiffusion cannot transfer and requires optimization for each individual image. NoiseCLR has the second-best yet lower transfer success rate, while other methods exhibit worse transferability.

- **Theoretically-grounded and supervision-free.** LOCO Edit is theoretically grounded. Besides, it is supervision-free, thus integrating no biases from other modules such as CLIP [36]. [37] shows CLIP sometimes can't capture detailed semantics such as color. We can observe failures in capturing detailed semantics for methods that utilize CLIP guidance such as BlendedDiffusion and Asyrp in Figure 6, where there are no edits or wrong edits.

# 6 Conclusion

We proposed a new low-rank controllable image editing method, LOCO Edit, which enables precise, one-step, localized editing using diffusion models. Our approach stems from the discovery of the locally linear posterior mean estimator in diffusion models and the identification of a low-dimensional semantic subspace in its Jacobian, theoretically verified under certain data assumptions. The identified editing directions possess several beneficial properties, such as linearity, homogeneity, and composability. Additionally, our method is versatile across different datasets and models and is applicable to text-supervised editing in T2I diffusion models. Through various experiments, we demonstrate the superiority of our method compared to existing approaches.

## Acknowledgement

We acknowledge support from NSF CAREER CCF-2143904, NSF CCF-2212066, NSF CCF-2212326, NSF IIS 2312842, NSF IIS 2402950, ONR N00014-22-1-2529, a gift grant from KLA, an Amazon AWS AI Award, MICDE Catalyst Grant. The authors acknowledge valuable discussions with Mr. Zekai Zhang (U. Michigan), Dr. Ismail R. Alkhouri (U. Michigan and MSU), Mr. Jinfan Zhou (U. Michigan), and Mr. Xiao Li (U. Michigan).

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

# Appendix

## A  Future Direction

We identify several future directions and limitations of the current work. The current theoretical framework explains mainly the unsupervised image editing part. A more solid and thorough analysis of text-supervised image editing is of significant importance in understanding T2I diffusion models, which is yet a difficult open problem in the field. For example, there is still a lack of geometric analysis of the relationship between subspaces under different text-prompt conditions [4, 19, 38, 66]. Based on such understandings, it may be possible to further discover benign properties of editing directions in T2I diffusion models, or design more efficient fine-tuning [67, 68] accordingly. Besides, the current method has the potential to be extended for combining coarse to fine editing across different time steps. Furthermore, it is worth exploring the direct manipulation of semantic spaces in flow-matching diffusion models and transformer-architecture diffusion models. Lastly, it is possible to connect the current finding to image or video representation learning in diffusion models [69, 70, 71, 72], extend to 3D editing of pose or shape [73, 74], or utilize the low-rank structures to build dictionaries [75].

## B  Discussion on Related Works

**Study of Latent Semantic Space in Generative Models.**   Although diffusion models have demonstrated their strengths in state-of-the-art image synthesis, the understanding of diffusion models is still far behind the other generative models such as Generative Adversarial Networks (GAN) [76, 57], the understanding of which can provide tools as well as inspiration for the understanding of diffusion models. Some recent works have identified such gaps, discovered latent semantic spaces in diffusion models [29], and further studied the properties of the latent space from a geometrical perspective [30]. These prior arts deepen our understanding of the latent semantic space in diffusion models, and inspire later works to study the structures of information represented in diffusion models from various angles. However, their semantic space is constrained to diffusion models using UNet architecture, and can not represent localized semantics. Our work explores an alternative space to study the semantic expression in diffusion models, inspired by our observation of the low-rank and locally linear Jacobian of the denoiser over the noisy images. We provide a theoretical framework for demonstrating and understanding such properties, which can deepen the interpretation of the learned data distribution in diffusion models.

**Image Editing in Unconditional Diffusion Models.**   Recent research has significantly improved the understanding of latent semantic spaces in diffusion models, enabling global image editing through either training-free methods [29, 30, 31] or by incorporating an additional lightweight model [30, 77]. However, these methods result in poor performance for localized edit. In contrast, our approach achieves localized editing without requiring supervised training. For localized edits, [25] builds on [30], enabling local edits by altering the intermediate layers of UNet. However, these approaches are restricted to UNet-based architectures in diffusion models and have largely ignored intrinsic properties like linearity and low-rankness. In comparison, our work provides a rigorous theoretical analysis of low-rankness and local linearity in diffusion models, and we are the first to offer a principled justification of the semantic significance of the basis used for editing. Moreover, our method is independent of specific network architectures.

Other recent works, such as [32], introduce training-free global audio and image editing based on a theoretical understanding of the posterior covariance matrix [33], also independent of UNet architectures. However, our approach offers a distinct perspective, providing complementary insights and new findings. We explore the low-rank nature and local linearity in PMP, offering rigorous theoretical analyses. Based on this, our proposed LOCO Edit method allows unsupervised and localized editing, which enables several advantageous properties including transferability, composability, and linearity – benign features that have not been explored in prior work. Further, we extend the method to unsupervised and text-supervised editing in various text-to-image models. Additionally, while [24] supports localized editing, it requires supervision from CLIP, lacks a theoretical basis, and is time-consuming for editing each image. In contrast, our method is more efficient, theoretically grounded, and free from failures or biases in CLIP. The CLIP-supervised may also exhibit a bias

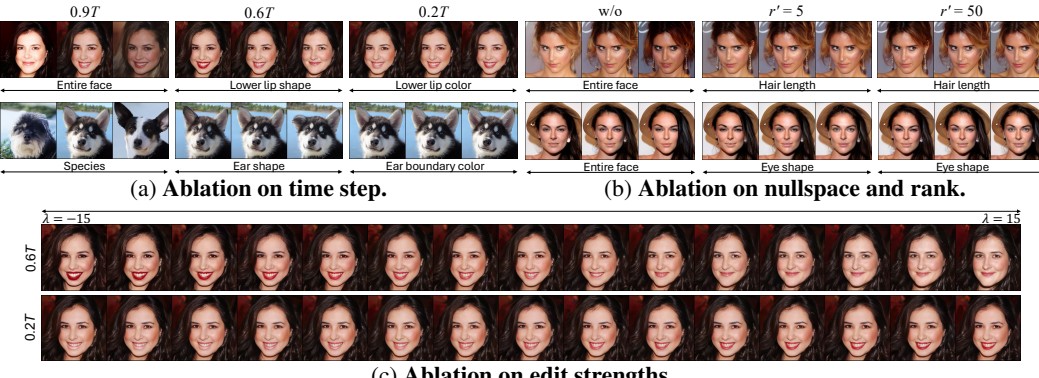

| 0.9T | 0.6T | 0.2T | w/o | $r' = 5$ | $r' = 50$ |

(a) **Ablation on time step.** (b) **Ablation on nullspace and rank.**

$\lambda = -15$        $\lambda = 15$

(c) **Ablation on edit strengths.**

Figure 7: **Ablation Study.** (a) Effects of one-step edit time. (b)Effects of using nullspace projection and rank. (c)Effects of editing strengths.

toward the CLIP score, leading to suboptimal editing results, as shown in Figure 6. In comparison, our method consistently enables high-quality edits without such bias.

**Image Editing in T2I Diffusion Models.** T2I image editing usually requires much more complicated sampling and training procedures, such as providing certainly learned guidance in the reverse sampling process [11], training an extra neural network [21], or fine-tuning the models for certain attributes [22]. Although effective, these methods often require extra training or even human intervention. Some other T2I image editing methods are training-free [46, 27, 28], and further enable editing with identifying masks [46], or optimizing the soft combination of text prompts [28]. These methods involve a continuous injection of the edit prompt during the generation process to gradually refine the generated image to have the target semantics. Though effective, all of the above methods (either training-free or not) as well as instruction-guided ones [78, 79, 80, 81] lack clear mathematical interpretations and requires text supervision. [23] discovers editing directions in T2I diffusion models through contrastive learning without text supervision, but is not generalizable to editing with text supervision. [30] has some theoretical basis and extends to an editing approach in T2I diffusion models with text supervision, but such supervision is only for unconditional sampling. In contrast, our extended T-LOCO Edit, which originated from the understanding of diffusion models, is the first method exploring single-step editing with or without text supervision for conditional sampling.

## C More Experiment Results on LOCO-Edit

### C.1 Ablation Studies

We conduct several important ablation studies on noise levels, the rank of nullspace projection, and editing strength, which demonstrates the robustness of our method.

- **Noise levels (i.e., editing time step $t$).** We conducted an ablation study on different noise levels, with representative examples shown in Figure 7a. The key observations are summarized as follows: (a) Larger noise levels (i.e., edit on $x_t$ with larger $t$) perform more coarse edit while small noise levels perform finer edit; (b) LOCO Edit is applicable to a generally large range of noise levels ([0.2T, 0.7T]) for precise edit.

- **Rank of nullspace projection $r'$.** Ablation study on nullspace projection is in Figure 7b (definition of $r'$ is in Algorithm 1). We present the key observations: (a) the local edit ability with no nullspace projection is weaker than that with nullspace projection; (b) when conducting nullspace projection, an effective low-rank estimation with $r' = 5$ can already achieve good local edit results.

- **Editing strength $\lambda$.** The linearity with respect to editing strengths is visualized in Figure 7c, with the key observations in addition to linearity: LOCO Edit is applicable to a generally wide range of editing strengths ([-15, 15]) to achieve localized edit.

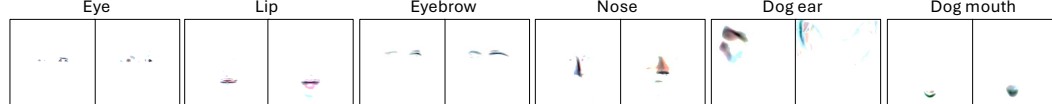

Figure 8: **Visualizing edit directions identified via LOCO Edit.** The edit directions are semantically meaningful.

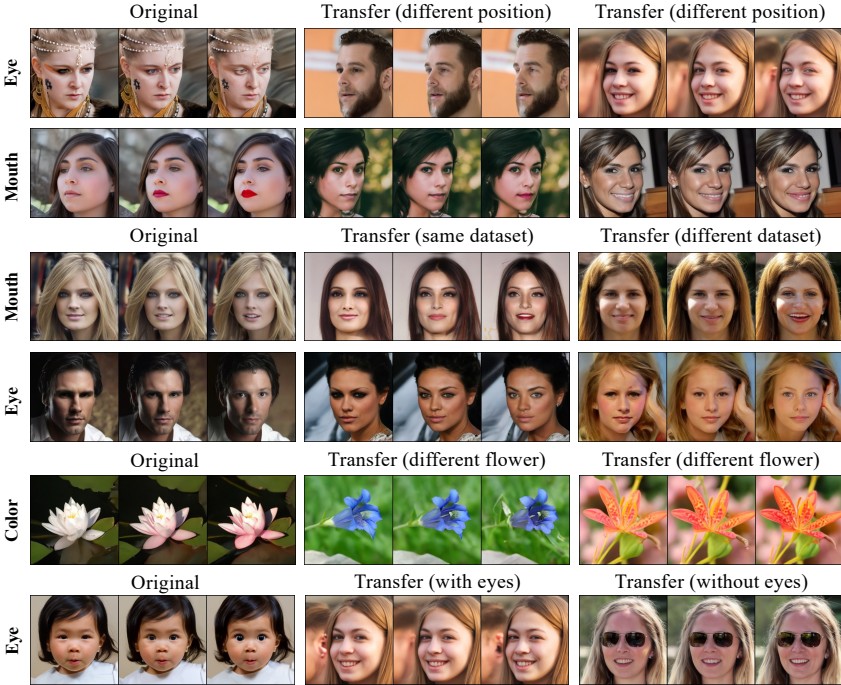

Figure 9: **Analyzing transferability of edit directions** to objects with different positions and shapes, images from different datasets, or images with no corresponding semantics.

## C.2 Visualization and Analysis of Editing Directions

We visualize the identified editing direction $v_p$ (see Algorithm 1) in Figure 8. The editing directions are semantically meaningful to the region of interest for editing. For example, the editing directions for eyes, lips, nose, etc., have similar shapes to eyes, lips, nose, etc.

Further, since the objects in datasets Flower, AFHQ, CelebA-HQ, and FFHQ are usually positioned at the center, the identified editing directions also tend to be at the center. Besides, objects could have different shapes, and semantics in some images do not exist in other images. To further study the robustness of transferability for the editing directions, we transfer editing directions to images with objects at different positions, from different datasets, with different shapes, and with no corresponding semantics. We present the results in Figure 9, with key observations that: (a) the edit directions are generally robust to gender differences, shape differences, moderate position differences, and dataset differences, illustrated in the first five rows of Figure 9 (b) transferring editing direction to images without corresponding semantics results in almost no editing (shown in the last row of Figure 9). Therefore, in practical applications, meaningful transfer editing scenarios for LOCO Edit occur when the transferred editing directions correspond to existing semantics in the target image (e.g., transferring the editing direction of "eyes" is effective only if the target image also contains eyes).

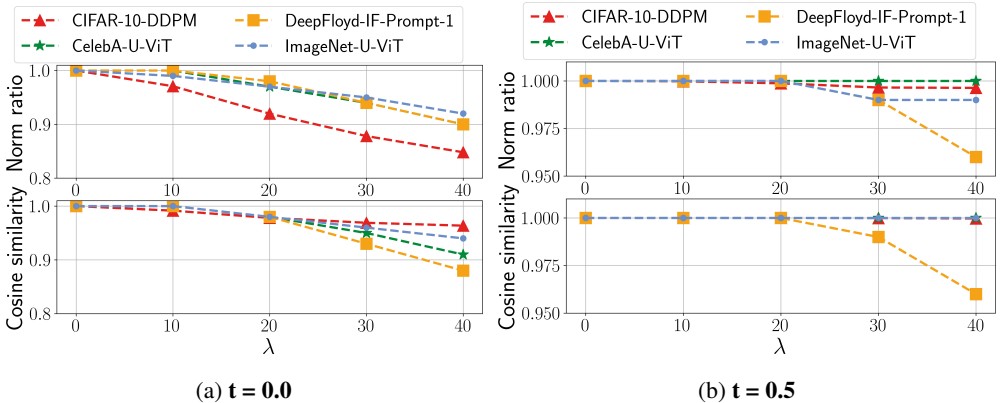

(a) **t = 0.0**                    (b) **t = 0.5**

Figure 10: **More results on the linearity of $f_{\boldsymbol{\theta},t}(\boldsymbol{x}_t, t)$.**

# D More Empirical Study on Low-rankness & Local Linearity

## D.1 Experiment Setup for Section 3.1

We evaluate the numerical rank of the denoiser function $\boldsymbol{x}_{\boldsymbol{\theta}}(\boldsymbol{x}_t, t)$ for DDPM (U-Net [49] architecture) on CIFAR-10 dataset [50] ($d = 32 \times 32 \times 3$), U-ViT [51] (Transformer based networks) on CelebA [52] ($d = 64 \times 64 \times 3$), ImageNet [53] datasets ($d = 64 \times 64 \times 3$) and DeepFloy IF [19] trained on LAION-5B [54] dataset ($d = 64 \times 64 \times 3$). Notably, U-ViT architecture uses the autoencoder to compress the image $\boldsymbol{x}_0$ to embedding vector $\boldsymbol{z}_0 = \texttt{Encoder}(\boldsymbol{x}_0)$, and adding noise to $\boldsymbol{z}_t$ for the diffusion forward process; and the reverse process replaces $\boldsymbol{x}_t, \boldsymbol{x}_{t-\Delta t}$ with $\boldsymbol{z}_t, \boldsymbol{z}_{t-\Delta t}$ in Equation (1). And the generated image $\boldsymbol{x}_0 = \texttt{Decoder}(\boldsymbol{z}_0)$. The PMP defined for U-ViT is:

$$\hat{\boldsymbol{x}}_{0,t} = f_{\boldsymbol{\theta},t}(\boldsymbol{z}_t; t) := \texttt{Decoder}\left(\frac{\boldsymbol{z}_t - \sqrt{1-\alpha_t}\boldsymbol{\epsilon}_{\boldsymbol{\theta}}(\boldsymbol{z}_t, t)}{\sqrt{\alpha_t}}\right). \tag{9}$$

The $\boldsymbol{J}_{\boldsymbol{\theta},t}(\boldsymbol{z}_t; t) = \nabla_{\boldsymbol{z}_t} f_{\boldsymbol{\theta},t}(\boldsymbol{z}_t; t)$ for $f_{\boldsymbol{\theta},t}(\boldsymbol{z}_t; t)$ defined above. For DeepFloy IF, there are three diffusion models, one for generation and the other two for super-resolution. Here we only evaluate $\boldsymbol{J}_{\boldsymbol{\theta},t}(\boldsymbol{z}_t; t)$ for diffusion generating the images.

Given a random initial noise $\boldsymbol{x}_T$, diffusion model $\boldsymbol{x}_{\boldsymbol{\theta}}$ generate image sequence $\{\boldsymbol{x}_t\}$ follows reverse sampler Equation (1). Along the sampling trajectory $\{\boldsymbol{x}_t\}$, for each $\boldsymbol{x}_t$, we calculate $\boldsymbol{J}_{\boldsymbol{\theta},t}(\boldsymbol{z}_t; t)$ and compute its numerical rank via

$$\widetilde{\texttt{rank}}(\boldsymbol{J}_{\boldsymbol{\theta},t}(\boldsymbol{x}_t)) = \arg\min_r \left\{ r : \frac{\sum_{i=1}^r \sigma_i^2 \left(\boldsymbol{J}_{\boldsymbol{\theta},t}(\boldsymbol{x}_t; t)\right)}{\sum_{i=1}^n \sigma_i^2 \left(\boldsymbol{J}_{\boldsymbol{\theta},t}(\boldsymbol{x}_t; t)\right)} > \eta^2 \right\}, \tag{10}$$

where $\sigma_i(\boldsymbol{A})$ denotes the $i$th largest singular value of $\boldsymbol{A}$. In our experiments, we set $\eta = 0.99$. We random generate 15 initialize noise $\boldsymbol{x}_t$ ($\boldsymbol{z}_t$ for U-ViT). We only use one prompt for DeepFloyd IF. We use DDIM with 100 steps for DDPM and DeepFloyd IF, DPM-Solver with 20 steps for U-ViT, and select some of the steps to calculate $\texttt{rank}(\boldsymbol{J}_{\boldsymbol{\theta},t}(\boldsymbol{x}_t; t))$, reported the averaged rank in Figure 2. To report the norm ratio and cosine similarity, we select the closest $t$ to 0.7 along the sampling trajectory and reported in Figure 2, i.e. $t = 0.71$ for DDPM, $t = 0.66$ for U-ViT and $t = 0.69$ for DeepFloyd IF. The norm ratio and cosine similarity are also averaged over 15 samples.

## D.2 More Experiments for Section 3.1

We illustrated the norm ratio and cosine similarity for more timesteps in Figure 10, more text prompts, and flow-matching-based diffusion model in Figure 11. More specifically, for the plot of $t = 0.0$, we exactly use $t = 0.04$ for DDPM, $t = 0.005$ for U-ViT and $t = 0.09$ for DeepFloyd IF; for the plot of $t = 0.5$, we exactly use $t = 0.49$ for DDPM, $t = 0.50$ for U-ViT and $t = 0.49$ for DeepFloyd IF. The results aligned with our results in Theorem 1 that when $t$ is closer the 1, the linearity of $f_{\boldsymbol{\theta},t}(\boldsymbol{x}_t, t)$ is better.

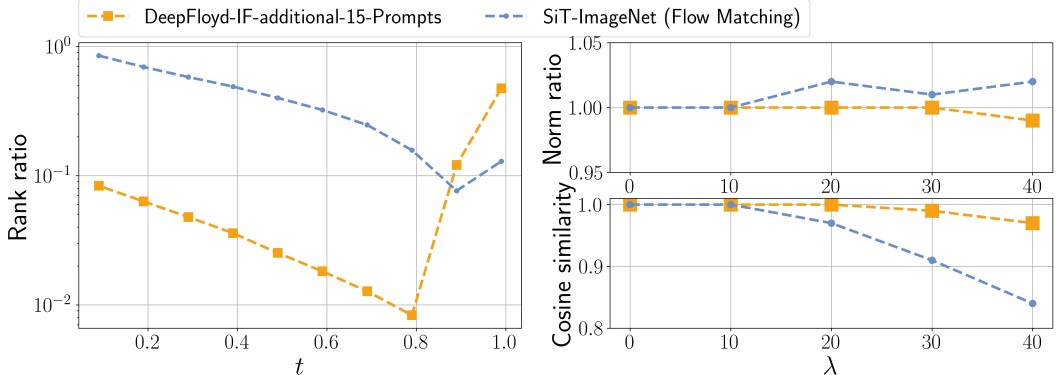

Figure 11: **More empirical study on low-rankness and local linearity on more prompts and models trained with flow-matching objectives.**

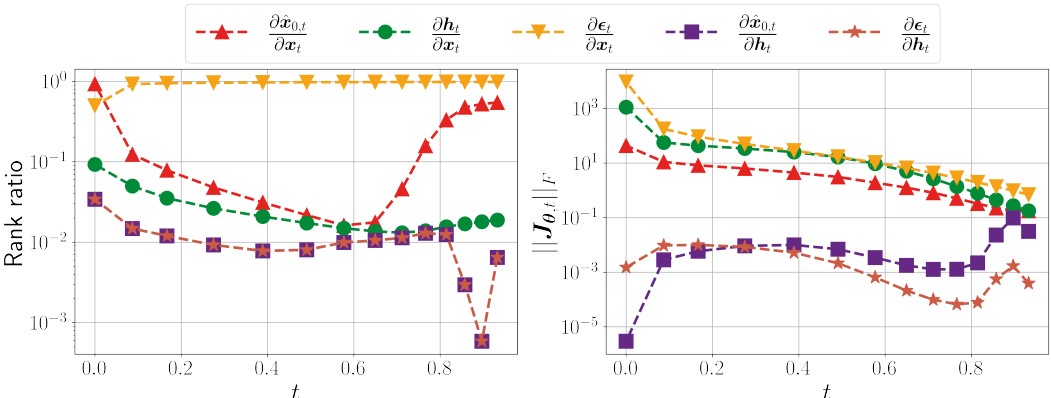

Figure 12: (Left) **Numerical rank of different jacobian $J$ at different timestep $t$.** (Right) **Frobenius norm of different jacobian $J$ at different timestep $t$**

### D.3 Comparison for Low-rankness & Local Linearity for Different Manifold

This section is an extension of Section 3.1. We study the low rankness and local linearity of more mappings between spaces of diffusion models. The sampling process of diffusion model involved the following space: $\boldsymbol{x}_t \in \mathcal{X}_t$, $\hat{\boldsymbol{x}}_{0,t} \in \mathcal{X}_{0,t}$, $\boldsymbol{h}_t \in \mathcal{H}_t$, $\boldsymbol{\epsilon}_t \in \mathcal{E}_t$, where $\mathcal{H}_t$ is the h-space of U-Net's bottleneck feature space [29] and $\mathcal{E}_t$ is the predict noise space. First, we explore the rank ratio of Jacobian $\boldsymbol{J}_{\boldsymbol{\theta},t}$ and Frobenius norm $||\boldsymbol{J}_{\boldsymbol{\theta},t}||_F$ for: $\frac{\partial \boldsymbol{h}_t}{\partial \boldsymbol{x}_t}, \frac{\partial \boldsymbol{\epsilon}_t}{\partial \boldsymbol{h}_t}, \frac{\partial \hat{\boldsymbol{x}}_{0,t}}{\partial \boldsymbol{h}_t}, \frac{\partial \boldsymbol{\epsilon}_t}{\partial \boldsymbol{x}_t}, \frac{\partial \hat{\boldsymbol{x}}_{0,t}}{\partial \boldsymbol{x}_t}$. We use DDPM with U-Net architecture, trained on CIFAR-10 dataset, and other experiment settings are the same as Appendix D.1, results are shown in Figure 12. The conclusion could be summarized as :

- $\frac{\partial \boldsymbol{h}_t}{\partial \boldsymbol{x}_t}, \frac{\partial \boldsymbol{\epsilon}_t}{\partial \boldsymbol{h}_t}, \frac{\partial \hat{\boldsymbol{x}}_{0,t}}{\partial \boldsymbol{h}_t}, \frac{\partial \hat{\boldsymbol{x}}_{0,t}}{\partial \boldsymbol{x}_t}$ *are low rank jacobian when* $t \in [0.2, 0.7]$. As shown in the left of Figure 12, rank ratio for $\frac{\partial \boldsymbol{h}_t}{\partial \boldsymbol{x}_t}, \frac{\partial \boldsymbol{\epsilon}_t}{\partial \boldsymbol{h}_t}, \frac{\partial \hat{\boldsymbol{x}}_{0,t}}{\partial \boldsymbol{h}_t}, \frac{\partial \hat{\boldsymbol{x}}_{0,t}}{\partial \boldsymbol{x}_t}$ is less than 0.1. It should be noted that:

  - $\widetilde{\text{rank}}(\frac{\partial \boldsymbol{\epsilon}_t}{\partial \boldsymbol{x}_t}) \geq d - \widetilde{\text{rank}}(\frac{\partial \hat{\boldsymbol{x}}_{0,t}}{\partial \boldsymbol{x}_t})$. This is because

  $$\widetilde{\text{rank}}(\frac{\sqrt{1-\alpha_t}}{\sqrt{\alpha_t}} \frac{\partial \boldsymbol{\epsilon}_t}{\partial \boldsymbol{x}_t}) \geq \widetilde{\text{rank}}(\frac{1}{\sqrt{\alpha_t}} \boldsymbol{I}_d) - \widetilde{\text{rank}}(\frac{\partial \hat{\boldsymbol{x}}_{0,t}}{\partial \boldsymbol{x}_t}).$$

  Therefore, $\frac{\partial \boldsymbol{\epsilon}_t}{\partial \boldsymbol{x}_t}$ is high rank when $\frac{\partial \hat{\boldsymbol{x}}_{0,t}}{\partial \boldsymbol{x}_t}$ is low rank.

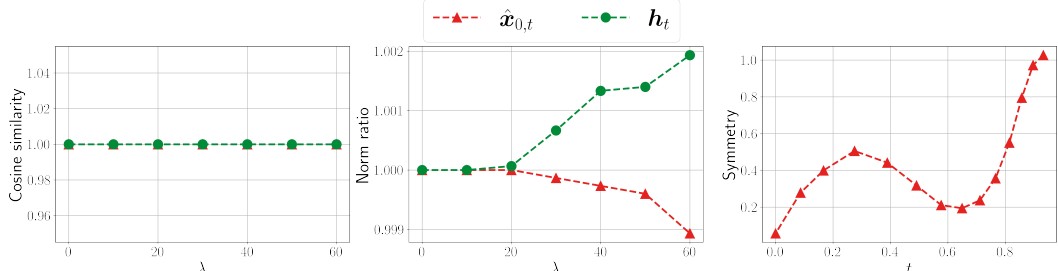

Figure 13: (Left, Middle) **Cosine similarity and norm ration of different mappings with respect to $\lambda$.** (Right) **Symmetric property of** $\dfrac{\partial x\hat{}_{0,t}}{\partial x_t}$ **with respect to timestep** $t$.

- $\widetilde{\text{rank}}(\dfrac{\partial \hat{x}_{0,t}}{\partial h_t}) = \widetilde{\text{rank}}(\dfrac{\partial \hat{x}_{0,t}}{\partial x_t})$ This is because $\hat{x}_{0,t} = \dfrac{x_t - \sqrt{1-\alpha_t}\epsilon_\theta(x_t, t)}{\sqrt{\alpha_t}}$ and $\dfrac{\partial x_t}{\partial h_t} = 0$

- *When $x_t$ fixed, $\hat{x}_{0,t}, \epsilon_t$ will change little when changing $h_t$.* As shown in the right of Figure 12, $||\dfrac{\partial \hat{x}_{0,t}}{\partial h_t}||_F \ll ||\dfrac{\partial \hat{x}_{0,t}}{\partial x_t}||_F$ and $\dfrac{\partial \epsilon_t}{\partial h_t} \ll \dfrac{\partial \epsilon_t}{\partial x_t}$. This means when $x_t$ fixed, $\hat{x}_{0,t}, \epsilon_t$ will change little when changing $h_t$.

Then, we also study the linearity of $h_t$ and $\hat{x}_{0,t}$ given $x_t$, using DDPM with U-Net architecture trained on CIFAR-10 dataset. We change the step size $\lambda$ defined in Equation (4). Results are shown in Figure 13, *both $h_t$ and $\hat{x}_{0,t}$ have good linearity with respect to $x_t$.*.

In Theorem 1, the jacobian $\nabla_{x_t}\mathbb{E}[x_0|x_t]$ is a symmetric matrix. Therefore, we also verify the symmetry of the jacobian over the PMP $J_{\theta,t}$. We use DDPM with U-Net architecture trained on CIFAR-10 dataset. At different timestep $t$, we measure $||J_{\theta,t} - J_{\theta,t}^\top||_F$. Results are shown on the right of Figure 13. $J_{\theta,t}$ has good symmetric property when $t < 0.1$ and $t \in [0.6, 0.7]$. Additionally, $J_{\theta,t}$ is low rank when $t \in [0.6, 0.7]$. So $J_{\theta,t}$ aligned with Theorem 1 $t \in [0.6, 0.7]$.

To the end, we want to based on the experiments in Figure 12 and Figure 13 to select the best space for out image editing method. $\dfrac{\partial \epsilon_t}{\partial x_t}$ is the high-rank matrix, not suitable for efficiently estimate the nullspace; $\dfrac{\partial \epsilon_t}{\partial h_t}$ and $\dfrac{\partial \hat{x}_{0,t}}{\partial h_t}$ has too small Frobenius norm to edit the image. Therefore, only $\dfrac{\partial h_t}{\partial x_t}$ and $\dfrac{\partial \hat{x}_{0,t}}{\partial x_t}$ are low-rank and linear for image editing. What's more, $h_t$ space is restricted to UNet architecture, but the property of the $\dfrac{\partial \hat{x}_{0,t}}{\partial x_t}$ does not depend on the UNet architecture and is verified in diffusion models using transformer architectures. Additionally, we could only apply masks on $\hat{x}_{0,t}$ but cannot on $h_t$. **Therefore, the PMP $f_{\theta,t}$ is the best mapping for image editing.**

# E Extra Details of LOCO Edit and T-LOCO Edit

## E.1 Generalized Power Method

The Generalized Power Method [34, 30] for calculating the op-$t$ singular vectors of the Jacobian is summarized in Algorithm 2. It efficiently computes the top-$k$ singular values and singular vectors of the Jacobian with a randomly initialized orthonormal $V \in \mathbb{R}^{d \times k}$.

## E.2 Unsupervised T-LOCO Edit

The overall method for DeepFloyd is summarized in Algorithm 3. For T2I diffusion models in the latent space such as Stable Diffusion and Latent Consistency Model, at time $t$, we additionally decode $\hat{z}_0$ into the image space $\hat{x}_0$ to enable masking and nullspace projection. The editing is still in the space of $z_t$.

---

**Algorithm 2** Generalized Power Method

---

1: **Input**: $\boldsymbol{f} : \mathbb{R}^d \to \mathbb{R}^d$, $\boldsymbol{x} \in \mathbb{R}^d$ and $\boldsymbol{V} \in \mathbb{R}^{d \times k}$

2: **Output**: $(\boldsymbol{U}, \boldsymbol{\Sigma}, \boldsymbol{V}^\top) - k$ top singular values and vectors of the Jacobian $\dfrac{\partial \boldsymbol{f}}{\partial \boldsymbol{x}}$

3: $\boldsymbol{y} \leftarrow \boldsymbol{f}(\boldsymbol{x})$

4: **if** $\boldsymbol{V}$ is empty **then**

5:     $\boldsymbol{V} \leftarrow$ i.i.d. standard Gaussian samples

6: **end if**

7: $\boldsymbol{Q}, \boldsymbol{R} \leftarrow \mathrm{QR}(\boldsymbol{V})$               ▷ Reduced QR decomposition

8: $\boldsymbol{V} \leftarrow \boldsymbol{Q}$                      ▷ Ensures $\boldsymbol{V}^\top \boldsymbol{V} = \boldsymbol{I}$

9: **while** stopping criteria **do**

10:     $\boldsymbol{U} \leftarrow \dfrac{\partial \boldsymbol{f}(\boldsymbol{x} + a\boldsymbol{V})}{\partial a}$ at $a = 0$          ▷ Batch forward

11:     $\hat{\boldsymbol{V}} \leftarrow \dfrac{\partial \left(\boldsymbol{U}^\top \boldsymbol{y}\right)}{\partial \boldsymbol{x}}$

12:     $\boldsymbol{V}, \boldsymbol{\Sigma}^2, \boldsymbol{R} \leftarrow \mathrm{SVD}(\hat{\boldsymbol{V}})$             ▷ Reduced SVD

13: **end while**

14: Orthonormalize $\boldsymbol{U}$

---

---

**Algorithm 3** Unsupervised T-LOCO Edit for T2I diffusion models

---

1: **Input**: Random noise $\boldsymbol{x}_T$, the mask $\Omega$, edit timestep $t$, pretrained diffusion model $\boldsymbol{\epsilon_\theta}$, editing scale $\lambda$, noise scheduler $\alpha_t, \sigma_t$, selected semantic index $k$, nullspace approximate rank $r$, original prompt $c_o$, null prompt $c_n$, classifier free guidance scale $s$.

2: **Output**: Edited image $\boldsymbol{x}_0'$,

3: $\boldsymbol{x}_t \leftarrow \mathtt{DDIM}(\boldsymbol{x}_T, 1, t, \boldsymbol{\epsilon_\theta}(\boldsymbol{x}_T, t, c_n) + s(\boldsymbol{\epsilon_\theta}(\boldsymbol{x}_T, t, c_o) - \boldsymbol{\epsilon_\theta}(\boldsymbol{x}_T, t, c_n)))$

4: $\hat{\boldsymbol{x}}_{0,t} \leftarrow \boldsymbol{f}_{\boldsymbol{\theta},t}^o(\boldsymbol{x}_t)$

5: Masking by $\tilde{\boldsymbol{x}}_{0,t} \leftarrow \mathcal{P}_\Omega(\hat{\boldsymbol{x}}_{0,t})$ and $\bar{\boldsymbol{x}}_{0,t} \leftarrow \hat{\boldsymbol{x}}_{0,t} - \tilde{\boldsymbol{x}}_{0,t}$     ▷ Use the mask for local image editing

6: The top-$k$ SVD $(\tilde{\boldsymbol{U}}_{t,k}, \tilde{\boldsymbol{\Sigma}}_{t,k}, \tilde{\boldsymbol{V}}_{t,k})$ of $\tilde{\boldsymbol{J}}_{\boldsymbol{\theta},t} = \dfrac{\partial \tilde{\boldsymbol{x}}_{0,t}}{\partial \boldsymbol{x}_t}$ ▷ Efficiently computed via generalized power method

7: The top-$r$ SVD $(\bar{\boldsymbol{U}}_{t,r}, \bar{\boldsymbol{\Sigma}}_{t,r}, \bar{\boldsymbol{V}}_{t,r})$ of $\bar{\boldsymbol{J}}_{\boldsymbol{\theta},t} = \dfrac{\partial \bar{\boldsymbol{x}}_{0,t}}{\partial \boldsymbol{x}_t}$ ▷ Efficiently computed via generalized power method

8: Pick direction $\boldsymbol{v} \leftarrow \tilde{\boldsymbol{V}}_{t,k}[:,i]$          ▷ Pick the $i^{th}$ singular vector for editing within the mask $\Omega$

9: Compute $\boldsymbol{v}_p \leftarrow (\boldsymbol{I} - \bar{\boldsymbol{V}}_{t,r}\bar{\boldsymbol{V}}_{t,r}^\top) \cdot \boldsymbol{v}$        ▷ Nullspace projection for editing within the mask $\Omega$

10: $\boldsymbol{v}_p \leftarrow \frac{\boldsymbol{v}_p}{\|\boldsymbol{v}_p\|_2}$                  ▷ Normalize the editing direction

11: $\boldsymbol{x}_t' \leftarrow \boldsymbol{x}_t + \lambda \boldsymbol{v}_p$

12: $\boldsymbol{x_0'} \leftarrow \mathtt{DDIM}(\boldsymbol{x}_t', t, 0, \boldsymbol{\epsilon_\theta}(\boldsymbol{x}_t, t, c_n) + s(\boldsymbol{\epsilon_\theta}(\boldsymbol{x}_t, t, c_o) - \boldsymbol{\epsilon_\theta}(\boldsymbol{x}_t, t, c_n)))$

---

### E.3 Text-suprvised T-LOCO Edit

Before introducing the algorithm, we define:

$$\boldsymbol{f}_{\boldsymbol{\theta},t}^o(\boldsymbol{x}_t) = \frac{\boldsymbol{x}_t - \alpha_t \sigma_t(\boldsymbol{\epsilon_\theta}(\boldsymbol{x}_t, t, c_n) + s(\boldsymbol{\epsilon_\theta}(\boldsymbol{x}_t, t, c_o) - \boldsymbol{\epsilon_\theta}(\boldsymbol{x}_t, t, c_n)))}{\alpha_t}, \tag{11}$$

and

$$\boldsymbol{f}_{\boldsymbol{\theta},t}^e(\boldsymbol{x}_t) = \boldsymbol{f}_{\boldsymbol{\theta},t}^o(\boldsymbol{x}_t) + \frac{m(\boldsymbol{\epsilon_\theta}(\boldsymbol{x}_t, t, c_e) - \boldsymbol{\epsilon_\theta}(\boldsymbol{x}_t, t, c_n)))}{\alpha_t}, \tag{12}$$

to be the posterior mean predictors when using classifier-free guidance on the original prompt $c_o$, and both the original prompt $c_o$ and the edit prompt $c_e$ accordingly.

**Algorithm.** The overall method for DeepFloyd is summarized in Algorithm 4. For T2I diffusion models in the latent space such as Stable Diffusion and Latent Consistency Model, at time $t$, we additionally decode $\hat{\boldsymbol{z}}_0$ into the image space $\hat{\boldsymbol{x}}_0$ to enable masking and nullspace projection. The editing is in the space of $\boldsymbol{z}_t$ for Stable Diffusion and Latent Consistency Model. The proposed method is not proposed as an approach beating other T2I editing methods, but as a way to both understand semantic correspondences in the low-rank subspaces of T2I diffusion models and utilize subspaces for semantic control in a more interpretable way. We hope to inspire and open up directions in understanding T2I diffusion models and utilize the understanding in versatile applications.

---

**Algorithm 4** Text-supervised T-LOCO Edit for T2I diffusion models

---

1: **Input**: Random noise $\boldsymbol{x}_T$, the mask $\Omega$,, edit timestep $t$, pretrained diffusion model $\boldsymbol{\epsilon}_{\boldsymbol{\theta}}$, editing scale $\lambda$, noise scheduler $\alpha_t, \sigma_t$, selected semantic index $k$, nullspace approximate rank $r$, original prompt $c_o$, edit prompt $c_e$, null prompt $c_n$, classifier free guidance scale $s$.

2: **Output**: Edited image $\boldsymbol{x}_0'$,

3: $\boldsymbol{x}_t \leftarrow \text{DDIM}(\boldsymbol{x}_T, 1, t, \boldsymbol{\epsilon}_{\boldsymbol{\theta}}(\boldsymbol{x}_T, t, c_n) + s(\boldsymbol{\epsilon}_{\boldsymbol{\theta}}(\boldsymbol{x}_T, t, c_o) - \boldsymbol{\epsilon}_{\boldsymbol{\theta}}(\boldsymbol{x}_T, t, c_n)))$

4: $\hat{\boldsymbol{x}}_{0,t}^o \leftarrow \boldsymbol{f}_{\boldsymbol{\theta},t}^o(\boldsymbol{x}_t)$

5: $\hat{\boldsymbol{x}}_{0,t}^e \leftarrow \boldsymbol{f}_{\boldsymbol{\theta},t}^e(\boldsymbol{x}_t)$

6: $\boldsymbol{d} \leftarrow \mathcal{P}_\Omega\left(\hat{\boldsymbol{x}}_{0,t}^e - \hat{\boldsymbol{x}}_{0,t}^o\right)$

7: $\tilde{\boldsymbol{x}}_{0,t} \leftarrow \mathcal{P}_\Omega(\hat{\boldsymbol{x}}_{0,t}^e)$

8: $\boldsymbol{v} \leftarrow \dfrac{\partial(\boldsymbol{d}^\top \tilde{\boldsymbol{x}}_{0,t})}{\partial \boldsymbol{x}_t}$ ▷ Get text-supervised editing direction within the mask

9: $\bar{\boldsymbol{x}}_{0,t} \leftarrow \hat{\boldsymbol{x}}_{0,t}^o - \mathcal{P}_\Omega(\hat{\boldsymbol{x}}_{0,t}^o)$

10: The top-$r$ SVD $(\bar{\boldsymbol{U}}_{t,r}, \bar{\boldsymbol{\Sigma}}_{t,r}, \bar{\boldsymbol{V}}_{t,r})$ of $\bar{\boldsymbol{J}}_{\boldsymbol{\theta},t} = \dfrac{\partial \bar{\boldsymbol{x}}_{0,t}}{\partial \boldsymbol{x}_t}$ ▷ Efficiently computed via generalized power method

11: $\boldsymbol{v}_p \leftarrow (\boldsymbol{I} - \bar{\boldsymbol{V}}_{t,r}\bar{\boldsymbol{V}}_{t,r}^\top) \cdot \boldsymbol{v}$ ▷ nullspace projection for editing within the mask

12: $\boldsymbol{v}_p \leftarrow \dfrac{\boldsymbol{v}_p}{\|\boldsymbol{v}_p\|_2}$ ▷ Normalize the editing direction

13: $\boldsymbol{x}_t' \leftarrow \boldsymbol{x}_t + \lambda \boldsymbol{v}_p$

14: $\boldsymbol{x}_0' \leftarrow \text{DDIM}(\boldsymbol{x}_t', t, 0, \boldsymbol{\epsilon}_{\boldsymbol{\theta}}(\boldsymbol{x}_t, t, c_n) + s(\boldsymbol{\epsilon}_{\boldsymbol{\theta}}(\boldsymbol{x}_t, t, c_o) - \boldsymbol{\epsilon}_{\boldsymbol{\theta}}(\boldsymbol{x}_t, t, c_n)))$

---

Here, we want to find a specific change direction $\boldsymbol{v}_p$ in the $\boldsymbol{x}_t$ space that can provide target edited images in the space of $\boldsymbol{x}_0$ by directly moving $\boldsymbol{x}_t$ along $\boldsymbol{v}_p$: the whole generation is not conditioned on $c_e$ at all, except that we utilize $c_e$ in finding the editing direction $\boldsymbol{v}_p$. This is in contrast to the method proposed in [30], where additional semantic information is injected via indirect x-space guidance conditioned on the edit prompt at time $t$. We hope to discover an editing direction that is expressive enough by itself to perform semantic editing.

**Intuition.** Let $\hat{\boldsymbol{x}}_{0,t}^o$ be the estimated posterior mean conditioned on the original prompt $c_o$, and $\hat{\boldsymbol{x}}_{0,t}^e$ be the estimated posterior mean conditioned on both the original prompt $c_o$ and the edit prompt $c_e$. Let $\boldsymbol{J}_{\boldsymbol{\theta},t}^o$ and $\boldsymbol{J}_{\boldsymbol{\theta},t}^e$ be their Jacobian over the noisy image $\boldsymbol{x}_t$ accordingly. The key intuition inspired by the unconditional cases are: i) the target editing direction $\boldsymbol{v}$ in the $\boldsymbol{x}_t$ space is homogeneous between the subspaces in $\boldsymbol{J}_{\boldsymbol{\theta},t}^o$ and $\boldsymbol{J}_{\boldsymbol{\theta},t}^e$; ii) the founded editing direction $\boldsymbol{v}$ can effectively reside in the direction of a right singular vector for both $\boldsymbol{J}_{\boldsymbol{\theta},t}^o$ and $\boldsymbol{J}_{\boldsymbol{\theta},t}^e$; iii) $\hat{\boldsymbol{x}}_{0,t}^e$ and $\hat{\boldsymbol{x}}_{0,t}^o$ are locally linear.

Define $\hat{\boldsymbol{x}}_{0,t}^e - \hat{\boldsymbol{x}}_{0,t}^o = \boldsymbol{d}$ as the change of estimated posterior mean. Let $\boldsymbol{J}_{\boldsymbol{\theta},t}^e = \boldsymbol{U}_t^e \boldsymbol{S}_t^e \boldsymbol{V}_t^{e^T}$, then $\boldsymbol{v} = \pm \boldsymbol{v}_i^e$ for some $i$. Besides, we have $\hat{\boldsymbol{x}}_{0,t}^e = \hat{\boldsymbol{x}}_{0,t}^e + \lambda^o \boldsymbol{J}_{\boldsymbol{\theta},t}^o \boldsymbol{v}$ and $\hat{\boldsymbol{x}}_{0,t}^o = \hat{\boldsymbol{x}}_{0,t}^e + \lambda^e \boldsymbol{J}_{\boldsymbol{\theta},t}^e \boldsymbol{v}$ due to homogeneity and linearity. Hence, $\boldsymbol{d} = -\lambda^e \boldsymbol{J}_{\boldsymbol{\theta},t}^e \boldsymbol{v} = \pm \lambda^e s_i^e \boldsymbol{u}_i^e$ and then $\boldsymbol{J}_{\boldsymbol{\theta},t}^{e^T} \boldsymbol{d} = \pm \lambda^e s_i^e s_i^e \boldsymbol{v}_i^e = \pm \lambda^e s_i^e s_i^e \boldsymbol{v}$, which is along the desired direction $\boldsymbol{v}$. And this $\boldsymbol{v}$ identified through the subspace in $\boldsymbol{J}_{\boldsymbol{\theta},t}^e$ can be effectively transferred in $\boldsymbol{J}_{\boldsymbol{\theta},t}^o$ for controlling the editing of target semantics. We further apply nullspace projection based on $\boldsymbol{J}_{\boldsymbol{\theta},t}^o$ to obtain the final editing direction $\boldsymbol{v}_p$.

# F   Proofs in Section 4

## F.1   Proofs of Lemma 1

*Proof of Lemma 1.* Under the Assumption 1, we could calculate the noised distribution $p_t(\boldsymbol{x}_t)$ at any timestep $t$,

$$p_t(\boldsymbol{x}_t) = \frac{1}{K} \sum_{k=1}^K p_t(\boldsymbol{x}_t|\text{"}\boldsymbol{x}_0 \text{ belongs to class } k\text{"})$$

$$= \frac{1}{K} \sum_{k=1}^K \int p_t(\boldsymbol{x}_t|\boldsymbol{x}_0 = \boldsymbol{M}_k \boldsymbol{a}_k, \text{"}\boldsymbol{x}_0 \text{ belongs to class } k\text{"}) \mathcal{N}(\boldsymbol{a}_k; \boldsymbol{0}, \boldsymbol{I}_{r_k}) d\boldsymbol{a}_k.$$

Because $a_k \sim \mathcal{N}(\mathbf{0}, \boldsymbol{I}_{r_k})$, $p_t(\boldsymbol{x}_t|\boldsymbol{x}_0 = \boldsymbol{M}_k\boldsymbol{a}_k$, "$\boldsymbol{x}_0$ belongs to class $k$") $\sim \mathcal{N}(\sqrt{\alpha_t}\boldsymbol{M}_k\boldsymbol{a}_k, (1 - \alpha_t)\boldsymbol{I}_d)$. From the relationship between conditional Gaussian distribution and marginal Gaussian distribution, it is easy to show that $p_t(\boldsymbol{x}_t|$"$\boldsymbol{x}_0$ belongs to class $k$") $\sim \mathcal{N}(\mathbf{0}, \alpha_t\boldsymbol{M}_k\boldsymbol{M}_k^\top + (1-\alpha_t)\boldsymbol{I}_d)$

Then, we have

$$p_t(\boldsymbol{x}_t) = \frac{1}{K}\sum_{k=1}^{K}\mathcal{N}(\mathbf{0}, \alpha_t\boldsymbol{M}_k\boldsymbol{M}_k^\top + (1-\alpha_t)\boldsymbol{I}_d).$$

Next, we compute the score function as follows:

$$
\begin{aligned}
\nabla_{\boldsymbol{x}_t}\mathrm{log}p_t(\boldsymbol{x}_t) &= \frac{\nabla_{\boldsymbol{x}_t}p_t(\boldsymbol{x}_t)}{p_t(\boldsymbol{x}_t)} \\
&= \frac{\sum_{k=1}^{K}\mathcal{N}(\mathbf{0}, \alpha_t\boldsymbol{M}_k\boldsymbol{M}_k^\top + (1-\alpha_t)\boldsymbol{I}_d)\left(-\frac{1}{1-\alpha_t}\boldsymbol{x}_t + \frac{\alpha_t}{1-\alpha_t}\boldsymbol{M}_k\boldsymbol{M}_k^\top\boldsymbol{x}_t\right)}{\sum_{k=1}^{K}\mathcal{N}(\mathbf{0}, \alpha_t\boldsymbol{M}_k\boldsymbol{M}_k^\top + (1-\alpha_t)\boldsymbol{I}_d)} \\
&= -\frac{1}{1-\alpha_t}\boldsymbol{x}_t + \frac{\alpha_t}{1-\alpha_t}\frac{\sum_{k=1}^{K}\mathcal{N}(\mathbf{0}, \alpha_t\boldsymbol{M}_k\boldsymbol{M}_k^\top + (1-\alpha_t)\boldsymbol{I}_d)\boldsymbol{M}_k\boldsymbol{M}_k^\top\boldsymbol{x}_t}{\sum_{k=1}^{K}\mathcal{N}(\mathbf{0}, \alpha_t\boldsymbol{M}_k\boldsymbol{M}_k^\top + (1-\alpha_t)\boldsymbol{I}_d)}.
\end{aligned}
$$

Based on Tweedie's formula [45, 82], the relationship between the score function and posterior is

$$\mathbb{E}[\boldsymbol{x}_0|\boldsymbol{x}_t] = \frac{\boldsymbol{x}_t + (1-\alpha_t)\nabla_{\boldsymbol{x}_t}\mathrm{log}p_t(\boldsymbol{x}_t)}{\sqrt{\alpha_t}}. \tag{13}$$

Therefore, the posterior mean is

$$
\begin{aligned}
\mathbb{E}[\boldsymbol{x}_0|\boldsymbol{x}_t] &= \sqrt{\alpha_t}\frac{\sum_{k=1}^{K}\mathcal{N}(\mathbf{0}, \alpha_t\boldsymbol{M}_k\boldsymbol{M}_k^\top + (1-\alpha_t)\boldsymbol{I}_d)\boldsymbol{M}_k\boldsymbol{M}_k^\top\boldsymbol{x}_t}{\sum_{k=1}^{K}\mathcal{N}(\mathbf{0}, \alpha_t\boldsymbol{M}_k\boldsymbol{M}_k^\top + (1-\alpha_t)\boldsymbol{I}_d)} \\
&= \sqrt{\alpha_t}\frac{\sum_{k=1}^{K}\exp\left(-\frac{1}{2}\boldsymbol{x}_t^\top\left(\alpha_t\boldsymbol{M}_k\boldsymbol{M}_k^\top + (1-\alpha_t)\boldsymbol{I}_d\right)^{-1}\boldsymbol{x}_t\right)\boldsymbol{M}_k\boldsymbol{M}_k^\top\boldsymbol{x}_t}{\sum_{k=1}^{K}\exp\left(-\frac{1}{2}\boldsymbol{x}_t^\top\left(\alpha_t\boldsymbol{M}_k\boldsymbol{M}_k^\top + (1-\alpha_t)\boldsymbol{I}_d\right)^{-1}\boldsymbol{x}_t\right)} \\
&= \sqrt{\alpha_t}\frac{\sum_{k=1}^{K}\exp\left(-\frac{1}{2(1-\alpha_t)}\left(\|\boldsymbol{x}_t\|^2 - \alpha_t\|\boldsymbol{M}_k^\top\boldsymbol{x}_t\|^2\right)\right)\boldsymbol{M}_k\boldsymbol{M}_k^\top\boldsymbol{x}_t}{\sum_{k=1}^{K}\exp\left(-\frac{1}{2(1-\alpha_t)}\left(\|\boldsymbol{x}_t\|^2 - \alpha_t\|\boldsymbol{M}_k^\top\boldsymbol{x}_t\|^2\right)\right)} \\
&= \sqrt{\alpha_t}\frac{\sum_{k=1}^{K}\exp\left(\frac{\alpha_t}{2(1-\alpha_t)}\|\boldsymbol{M}_k^\top\boldsymbol{x}\|^2\right)\boldsymbol{M}_k\boldsymbol{M}_k^\top\boldsymbol{x}_t}{\sum_{k=1}^{K}\exp\left(\frac{\alpha_t}{2(1-\alpha_t)}\|\boldsymbol{M}_k^\top\boldsymbol{x}\|^2\right)},
\end{aligned}
$$

where the third equation is obtained by Woodbury formula [83] $(\alpha_t\boldsymbol{M}_k\boldsymbol{M}_k^\top + (1-\alpha_t)\boldsymbol{I}_d)^{-1} = \frac{1}{1-\alpha_t}\left(\boldsymbol{I}_d - \alpha_t\boldsymbol{M}_k\boldsymbol{M}_k^\top\right)$. $\qquad\square$

### F.2 Proofs of Theorem 1

**Lemma 2.** *The jacobian of the poster mean is*

$$\nabla_{\boldsymbol{x}_t}\mathbb{E}\left[\boldsymbol{x}_0|\boldsymbol{x}_t\right] = \sqrt{\alpha_t}\underbrace{\sum_{k=1}^{K}\omega_k(\boldsymbol{x}_t)\boldsymbol{M}_k\boldsymbol{M}_k^{\top}}_{\boldsymbol{A}:=}$$

$$+\frac{\alpha_t\sqrt{\alpha_t}}{(1-\alpha_t)}\underbrace{\sum_{k=1}^{K}\omega_k(\boldsymbol{x}_t)\boldsymbol{M}_k\boldsymbol{M}_k^{\top}\boldsymbol{x}_t\boldsymbol{x}_t^{\top}\boldsymbol{M}_k\boldsymbol{M}_k^{\top}}_{\boldsymbol{B}:=} \tag{14}$$

$$-\frac{\alpha_t\sqrt{\alpha_t}}{(1-\alpha_t)}\underbrace{\left(\sum_{k=1}^{K}\omega_k(\boldsymbol{x}_t)\boldsymbol{M}_k\boldsymbol{M}_k^{\top}\right)\boldsymbol{x}_t\boldsymbol{x}_t^{\top}\left(\sum_{k=1}^{K}\omega_k(\boldsymbol{x}_t)\boldsymbol{M}_k\boldsymbol{M}_k^{\top}\right)^{\top}}_{\boldsymbol{C}:=},$$

*where* $\omega_k(\boldsymbol{x}_t) := \dfrac{\exp\left(\dfrac{\alpha_t}{2\left(1-\alpha_t\right)}\|\boldsymbol{M}_k^{\top}\boldsymbol{x}_t\|^2\right)}{\sum_{l=1}^{K}\exp\left(\dfrac{\alpha_t}{2(1-\alpha_t)}\|\boldsymbol{M}_l^{\top}\boldsymbol{x}\|^2\right)}$

*Proof of Lemma 2.* Let $\omega_k(\boldsymbol{x}_t) := \dfrac{\exp\left(\dfrac{\alpha_t}{2\left(1-\alpha_t\right)}\|\boldsymbol{M}_k^{\top}\boldsymbol{x}_t\|^2\right)}{\sum_{l=1}^{K}\exp\left(\dfrac{\alpha_t}{2(1-\alpha_t)}\|\boldsymbol{M}_l^{\top}\boldsymbol{x}\|^2\right)}$, so we have:

$$\mathbb{E}\left[\boldsymbol{x}_0|\boldsymbol{x}_t\right] = \sqrt{\alpha_t}\sum_{k=1}^{K}\omega_k(\boldsymbol{x}_t)\boldsymbol{M}_k\boldsymbol{M}_k^{\top}\boldsymbol{x}_t$$

$$\nabla_{\boldsymbol{x}_t}\omega_k(\boldsymbol{x}_t) = \frac{\alpha_t}{(1-\alpha_t)}\omega_k(\boldsymbol{x}_t)\left[\boldsymbol{M}_k\boldsymbol{M}_k^{\top}\boldsymbol{x}_t - \sum_{l=1}^{K}\omega_l(\boldsymbol{x}_t)\boldsymbol{M}_l\boldsymbol{M}_l^{\top}\boldsymbol{x}_t\right]$$

So:

$$\nabla_{\boldsymbol{x}_t}\mathbb{E}\left[\boldsymbol{x}_0|\boldsymbol{x}_t\right] = \sqrt{\alpha_t}\sum_{k=1}^{K}\omega_k(\boldsymbol{x}_t)\boldsymbol{M}_k\boldsymbol{M}_k^{\top} + \sqrt{\alpha_t}\sum_{k=1}^{K}\nabla_{\boldsymbol{x}_t}\omega_k(\boldsymbol{x}_t)\boldsymbol{x}_t^{\top}\boldsymbol{M}_k\boldsymbol{M}_k^{\top}$$

$$= \sqrt{\alpha_t}\sum_{k=1}^{K}\omega_k(\boldsymbol{x}_t)\boldsymbol{M}_k\boldsymbol{M}_k^{\top}$$

$$+\frac{\alpha_t\sqrt{\alpha_t}}{(1-\alpha_t)}\sum_{k=1}^{K}\omega_k(\boldsymbol{x}_t)\boldsymbol{M}_k\boldsymbol{M}_k^{\top}\boldsymbol{x}_t\boldsymbol{x}_t^{\top}\boldsymbol{M}_k\boldsymbol{M}_k^{\top}$$

$$-\frac{\alpha_t\sqrt{\alpha_t}}{(1-\alpha_t)}\left(\sum_{k=1}^{K}\omega_k(\boldsymbol{x}_t)\boldsymbol{M}_k\boldsymbol{M}_k^{\top}\right)\boldsymbol{x}_t\boldsymbol{x}_t^{\top}\left(\sum_{k=1}^{K}\omega_k(\boldsymbol{x}_t)\boldsymbol{M}_k\boldsymbol{M}_k^{\top}\right)^{\top}.$$

$$\square$$

**Lemma 3.** *Assume second-order partial derivatives of $p_t(\boldsymbol{x}_t)$ exist for any $\boldsymbol{x}_t$, then the posterior mean $\nabla_{\boldsymbol{x}_t}\mathbb{E}\left[\boldsymbol{x}_0|\boldsymbol{x}_t\right]$ satisfied $\nabla_{\boldsymbol{x}_t}\mathbb{E}\left[\boldsymbol{x}_0|\boldsymbol{x}_t\right] = \nabla_{\boldsymbol{x}_t}\mathbb{E}^{\top}\left[\boldsymbol{x}_0|\boldsymbol{x}_t\right].$*

*Proof of Lemma 3.* By taking the gradient of Equation (13) with respect to $\boldsymbol{x}_t$ for both side, because the second-order partial derivatives of $p_t(\boldsymbol{x}_t)$ exist for any $\boldsymbol{x}_t$, we have:

$$\nabla_{\boldsymbol{x}_t}\mathbb{E}[\boldsymbol{x}_0|\boldsymbol{x}_t] = \frac{\boldsymbol{I} + (1-\alpha_t)\nabla_{\boldsymbol{x}_t}^2\log p_t(\boldsymbol{x}_t)}{\sqrt{\alpha_t}}.$$

The hessian of $\log p_t(\boldsymbol{x}_t)$ is symmetric, so we have:

$$\nabla_{\boldsymbol{x}_t}\mathbb{E}^\top[\boldsymbol{x}_0|\boldsymbol{x}_t] = \frac{\boldsymbol{I} + (1-\alpha_t)\left(\nabla^2_{\boldsymbol{x}_t}\log p_t(\boldsymbol{x}_t)\right)^\top}{\sqrt{\alpha_t}} = \frac{\boldsymbol{I} + (1-\alpha_t)\nabla^2_{\boldsymbol{x}_t}\log p_t(\boldsymbol{x}_t)}{\sqrt{\alpha_t}} = \nabla_{\boldsymbol{x}_t}\mathbb{E}[\boldsymbol{x}_0|\boldsymbol{x}_t].$$

Notably, the symmetric of $\nabla_{\boldsymbol{x}_t}\mathbb{E}[\boldsymbol{x}_0|\boldsymbol{x}_t]$ holds without the Assumption 1. $\qquad\square$

*Proof of Theorem 1.* **First, let's prove the low-rankness of the posterior mean**. From Lemma 2,

$$\nabla_{\boldsymbol{x}_t}\mathbb{E}\left[\boldsymbol{x}_0|\boldsymbol{x}_t\right] = \sqrt{\alpha_t}\boldsymbol{A} + \frac{\alpha_t\sqrt{\alpha_t}}{(1-\alpha_t)}\boldsymbol{B} - \frac{\alpha_t\sqrt{\alpha_t}}{(1-\alpha_t)}\boldsymbol{C}$$

$$= \sum_{k=1}^K \boldsymbol{M}_k\boldsymbol{M}_k^\top\left(\sqrt{\alpha_t}\boldsymbol{A} + \frac{\alpha_t\sqrt{\alpha_t}}{(1-\alpha_t)}\boldsymbol{B} - \frac{\alpha_t\sqrt{\alpha_t}}{(1-\alpha_t)}\boldsymbol{C}\right),$$

where the second equation is obtained due to the fact that $\sum_{k=1}^K \boldsymbol{M}_k\boldsymbol{M}_k^\top\boldsymbol{A} = \boldsymbol{A}, \sum_{k=1}^K \boldsymbol{M}_k\boldsymbol{M}_k^\top\boldsymbol{B} = \boldsymbol{B}, \sum_{k=1}^K \boldsymbol{M}_k\boldsymbol{M}_k^\top\boldsymbol{C} = \boldsymbol{C}$. Therefore, we have:

$$
\begin{aligned}
rank\left(\nabla_{\boldsymbol{x}_t}\mathbb{E}\left[\boldsymbol{x}_0|\boldsymbol{x}_t\right]\right) &= rank\left(\sum_{k=1}^K \boldsymbol{M}_k\boldsymbol{M}_k^\top\left(\sqrt{\alpha_t}\boldsymbol{A} + \frac{\alpha_t\sqrt{\alpha_t}}{(1-\alpha_t)}\boldsymbol{B} - \frac{\alpha_t\sqrt{\alpha_t}}{(1-\alpha_t)}\boldsymbol{C}\right)\right) \\
&\leq rank\left(\sum_{k=1}^K \boldsymbol{M}_k\boldsymbol{M}_k^\top\right) = \sum_{k=1}^K r_k
\end{aligned}
\tag{15}
$$

**Then, we prove the linearity**:

①: $||\mathbb{E}\left[\boldsymbol{x}_0|\boldsymbol{x}_t + \lambda\Delta\boldsymbol{x}\right] - \mathbb{E}\left[\boldsymbol{x}_0|\boldsymbol{x}_t\right] - \lambda\nabla_{\boldsymbol{x}_t}\mathbb{E}[\boldsymbol{x}_0|\boldsymbol{x}_t]\Delta\boldsymbol{x}||_2$

$=||\sqrt{\alpha_t}\sum_{k=1}^K \left(\omega_k(\boldsymbol{x}_t + \lambda\Delta\boldsymbol{x}) - \omega_k(\boldsymbol{x}_t)\right)\boldsymbol{M}_k\boldsymbol{M}_k^\top(\boldsymbol{x}_t + \lambda\Delta\boldsymbol{x}) - \lambda\sum_{k=1}^K \nabla_{\boldsymbol{x}_t}\omega_k(\boldsymbol{x}_t)\boldsymbol{x}_t^\top\boldsymbol{M}_k\boldsymbol{M}_k^\top\Delta\boldsymbol{x}||_2$

$=||\sqrt{\alpha_t}\sum_{k=1}^K \left(\lambda\nabla_{\boldsymbol{x}_t}^\top\omega_k(\boldsymbol{x}_t + \lambda_1\Delta\boldsymbol{x})\Delta\boldsymbol{x}\right)\boldsymbol{M}_k\boldsymbol{M}_k^\top(\boldsymbol{x}_t + \lambda\Delta\boldsymbol{x}) - \lambda\sum_{k=1}^K \nabla_{\boldsymbol{x}_t}\omega_k(\boldsymbol{x}_t)\boldsymbol{x}_t^\top\boldsymbol{M}_k\boldsymbol{M}_k^\top\Delta\boldsymbol{x}||_2$

$\leq\lambda\left(\sum_{k=1}^K \sqrt{\alpha_t}\nabla_{\boldsymbol{x}_t}^\top\omega_k(\boldsymbol{x}_t + \lambda_1\Delta\boldsymbol{x})\Delta\boldsymbol{x}||\boldsymbol{M}_k^\top(\boldsymbol{x}_t + \lambda\Delta\boldsymbol{x})||_2 + \boldsymbol{x}_t^\top\boldsymbol{M}_k\boldsymbol{M}_k^\top\Delta\boldsymbol{x}||\nabla_{\boldsymbol{x}_t}^\top\omega_k(\boldsymbol{x}_t)||_2\right)$

$\leq\lambda\sum_{k=1}^K \left(\sqrt{\alpha_t}||\nabla_{\boldsymbol{x}_t}\omega_k(\boldsymbol{x}_t + \lambda_1\Delta\boldsymbol{x})||_2||\boldsymbol{M}_k^\top(\boldsymbol{x}_t + \lambda\Delta\boldsymbol{x})||_2 + ||\nabla_{\boldsymbol{x}_t}\omega_k(\boldsymbol{x}_t)||_2||\boldsymbol{M}_k^\top\boldsymbol{x}_t||_2\right)$

where the first equation plug in the formula of $\nabla_{\boldsymbol{x}_t}\mathbb{E}\left[\boldsymbol{x}_0|\boldsymbol{x}_t\right] = \sqrt{\alpha_t}\sum_{k=1}^K \omega_k(\boldsymbol{x}_t)\boldsymbol{M}_k\boldsymbol{M}_k^\top + \sqrt{\alpha_t}\sum_{k=1}^K \nabla_{\boldsymbol{x}_t}\omega_k(\boldsymbol{x}_t)\boldsymbol{x}_t^\top\boldsymbol{M}_k\boldsymbol{M}_k^\top$ and the second equation use the mean value theorem $\omega_k(\boldsymbol{x}_t + \lambda\Delta\boldsymbol{x}) - \omega_k(\boldsymbol{x}_t) = \lambda\nabla_{\boldsymbol{x}_t}^\top\omega_k(\boldsymbol{x}_t + \lambda_1\Delta\boldsymbol{x})\Delta\boldsymbol{x}, \lambda_1 \in (0, \lambda)$.

$\textcircled{2} : ||\nabla_{\boldsymbol{x}_t}\omega_k(\boldsymbol{x}_t + \lambda_1 \Delta\boldsymbol{x})||_2$

$$= \frac{\alpha_t}{(1-\alpha_t)}\omega_k||\boldsymbol{M}_k\boldsymbol{M}_k^\top(\boldsymbol{x}_t + \lambda_1\Delta\boldsymbol{x}) - \sum_{l=1}^K \omega_l \boldsymbol{M}_l\boldsymbol{M}_l^\top(\boldsymbol{x}_t + \lambda_1\Delta\boldsymbol{x})||_2$$

$$\leq \frac{\alpha_t}{(1-\alpha_t)}\omega_k\left(||\boldsymbol{M}_k^\top\boldsymbol{x}_t||_2 + \sum_{l=1}^K \omega_l||\boldsymbol{M}_l^\top\boldsymbol{x}_t||_2 + \lambda_1||\boldsymbol{M}_k^\top\Delta\boldsymbol{x}||_2 + \lambda_1\sum_{l=1}^K\omega_l||\boldsymbol{M}_l^\top\Delta\boldsymbol{x}||_2\right)$$

$$\leq \frac{\alpha_t}{(1-\alpha_t)}\omega_k\left(||\boldsymbol{M}_k^\top||_F||\boldsymbol{x}_t||_2 + \sum_{l=1}^K \omega_l||\boldsymbol{M}_l^\top||_F||\boldsymbol{x}_t||_2 + \lambda_1||\boldsymbol{M}_k^\top||_F + \lambda_1\sum_{l=1}^K\omega_l||\boldsymbol{M}_l^\top||_F\right)$$

$$\leq \frac{\alpha_t}{(1-\alpha_t)}\omega_k\left(r_k + \sum_{l=1}^K\omega_l r_l\right)\left(\sqrt{2}\max\{||\boldsymbol{x}_0||_2, ||\boldsymbol{\epsilon}||_2\} + \lambda_1\right)$$

$$\leq \frac{\alpha_t}{(1-\alpha_t)}\omega_k(\boldsymbol{x}_t + \lambda_1\Delta\boldsymbol{x}) \cdot \underbrace{2\cdot\max_k r_k \cdot \left(\sqrt{2}\max\{||\boldsymbol{x}_0||_2, ||\boldsymbol{\epsilon}||_2\} + \lambda_1\right)}_{C_1 :=},$$

where the third inequality use the fact that $||\boldsymbol{x}_t||_2 = ||\sqrt{\alpha_t}\boldsymbol{x}_0 + \sqrt{1-\alpha_t}\boldsymbol{\epsilon}||_2 \leq ||\sqrt{\alpha_t}\boldsymbol{x}_0||_2 + ||\sqrt{1-\alpha_t}\boldsymbol{\epsilon}||_2 \leq \sqrt{2}\max\{||\boldsymbol{x}_0||_2, ||\boldsymbol{\epsilon}||_2\}$, we simplified $\omega_k(\boldsymbol{x}_t + \lambda_1\Delta\boldsymbol{x})$ as $\omega_k$ in this prove, and $C_1$ defined in the last inequality is independent of $t$. Similarly, we could prove that:

$$\textcircled{3} : ||\boldsymbol{M}_k\boldsymbol{M}_k^\top(\boldsymbol{x}_t + \lambda\Delta\boldsymbol{x})||_2 \leq \underbrace{\max_k r_k \cdot \left(\sqrt{2}\max\{||\boldsymbol{x}_0||_2, ||\boldsymbol{\epsilon}||_2\} + \lambda\right)}_{C_2 :=},$$

$$\textcircled{4} : ||\nabla_{\boldsymbol{x}_t}\omega_k(\boldsymbol{x}_t)||_2 \leq \frac{\alpha_t}{(1-\alpha_t)}\omega_k(\boldsymbol{x}_t)\underbrace{2\sqrt{2}\cdot\max_k r_k \cdot \max\{||\boldsymbol{x}_0||_2, ||\boldsymbol{\epsilon}||_2\}}_{C_3 :=},$$

$$\textcircled{5} : ||\boldsymbol{M}_k\boldsymbol{M}_k^\top\boldsymbol{x}_t||_2 \leq \underbrace{\sqrt{2}\max_k r_k \cdot \max\{||\boldsymbol{x}_0||_2, ||\boldsymbol{\epsilon}||_2\}}_{C_4 :=}.$$

Here, $C_1 = \mathcal{O}(\lambda), C_2 = \mathcal{O}(\lambda), C_3 = \mathcal{O}(\lambda), C_4 = \mathcal{O}(\lambda)$. After plugin $\textcircled{2}, \textcircled{3}, \textcircled{4}, \textcircled{5}$ to $\textcircled{1}$, we could obtain:

$$||\mathbb{E}[\boldsymbol{x}_0|\boldsymbol{x}_t + \lambda\Delta\boldsymbol{x}] - \mathbb{E}[\boldsymbol{x}_0|\boldsymbol{x}_t] - \lambda\nabla_{\boldsymbol{x}_t}\mathbb{E}[\boldsymbol{x}_0|\boldsymbol{x}_t]\Delta\boldsymbol{x}||_2$$

$$\leq \lambda\sqrt{\alpha_t}\sum_{k=1}^K \frac{\alpha_t}{(1-\alpha_t)}\omega_k(\boldsymbol{x}_t + \lambda_1\Delta\boldsymbol{x})C_1 C_2 + \lambda\sum_{k=1}^K \frac{\alpha_t}{(1-\alpha_t)}\omega_k(\boldsymbol{x}_t)C_3 C_4$$

$$= \lambda\frac{\alpha_t}{(1-\alpha_t)}\mathcal{O}(\lambda)$$

**Finally, let's prove the property of the left singular vector of $\nabla_{\boldsymbol{x}_t}\mathbb{E}[\boldsymbol{x}_0|\boldsymbol{x}_t]$:**

From Lemma 3, the eigenvalue decomposition of $\nabla_{\boldsymbol{x}_t}\mathbb{E}[\boldsymbol{x}_0|\boldsymbol{x}_t]$ could be written as $\nabla_{\boldsymbol{x}_t}\mathbb{E}[\boldsymbol{x}_0|\boldsymbol{x}_t] = \boldsymbol{U}_t\Lambda_t\boldsymbol{U}_t^\top$, where $\Lambda_t = \mathrm{diag}(\lambda_{t,1}, \ldots, \lambda_{t,r}, \ldots, 0)$, and the relation between eigenvalue decomposition and singular value decomposition of $\nabla_{\boldsymbol{x}_t}\mathbb{E}[\boldsymbol{x}_0|\boldsymbol{x}_t]$ could be summarized as for all $i \in [r]$:

$$\sigma_{t,i} = |\lambda_{t,i}|, \quad \boldsymbol{v}_i = \mathrm{sign}(\lambda_{t,i})\boldsymbol{u}_i,$$

where $\mathrm{sign}(\cdot)$ is the sign function. Therefore, we have:

$$\boldsymbol{U}_{t,1}\boldsymbol{U}_{t,1}^\top = \boldsymbol{V}_{t,1}\boldsymbol{V}_{t,1}^\top, \tag{16}$$

given $\boldsymbol{V}_{t,1} := [\boldsymbol{v}_{t,1}, \boldsymbol{v}_{t,2}, \ldots, \boldsymbol{v}_{t,r}]$. From Lemma 2, we define:

$$\nabla_{\boldsymbol{x}_t}\mathbb{E}[\boldsymbol{x}_0|\boldsymbol{x}_t] = \sqrt{\alpha_t}\sum_{k=1}^K \omega_k(\boldsymbol{x}_t)\boldsymbol{M}_k\boldsymbol{M}_k^\top + \underbrace{\sqrt{\alpha_t}\sum_{k=1}^K \nabla_{\boldsymbol{x}_t}\omega_k(\boldsymbol{x}_t)\boldsymbol{x}_t^\top\boldsymbol{M}_k\boldsymbol{M}_k^\top}_{\boldsymbol{\Delta}_t :=}.$$

From the full singular value decomposition of $\nabla_{\boldsymbol{x}_t} \mathbb{E}\left[\boldsymbol{x}_0 | \boldsymbol{x}_t\right]$ and $\sqrt{\alpha_t} \sum_{k=1}^K \omega_k(\boldsymbol{x}_t) \boldsymbol{M}_k \boldsymbol{M}_k^\top$:

$$\nabla_{\boldsymbol{x}_t} \mathbb{E}\left[\boldsymbol{x}_0 | \boldsymbol{x}_t\right] = \begin{bmatrix} \boldsymbol{U}_{t,1} & \boldsymbol{U}_{t,2} \end{bmatrix} \begin{bmatrix} \boldsymbol{\Sigma}_{t,1} & \boldsymbol{0} \\ \boldsymbol{0} & \boldsymbol{\Sigma}_{t,2} \end{bmatrix} \begin{bmatrix} \boldsymbol{V}_{t,1} \\ \boldsymbol{V}_{t,2} \end{bmatrix}^\top ,$$

$$\sqrt{\alpha_t} \sum_{k=1}^K \omega_k(\boldsymbol{x}_t) \boldsymbol{M}_k \boldsymbol{M}_k^\top = \begin{bmatrix} \hat{\boldsymbol{U}}_{t,1} & \hat{\boldsymbol{U}}_{t,2} \end{bmatrix} \begin{bmatrix} \hat{\boldsymbol{\Sigma}}_{t,1} & \boldsymbol{0} \\ \boldsymbol{0} & \hat{\boldsymbol{\Sigma}}_{t,2} \end{bmatrix} \begin{bmatrix} \hat{\boldsymbol{V}}_{t,1} \\ \hat{\boldsymbol{V}}_{t,2} \end{bmatrix}^\top .$$

where:

$$\boldsymbol{\Sigma}_{t,1} = \begin{bmatrix} \sigma_{t,1} & & \\ & \ddots & \\ & & \sigma_{t,r} \end{bmatrix}, \boldsymbol{\Sigma}_{t,2} = \begin{bmatrix} \sigma_{t,r+1} & & \\ & \ddots & \\ & & \sigma_{t,n} \end{bmatrix},$$

$$\hat{\boldsymbol{\Sigma}}_{t,1} = \begin{bmatrix} \hat{\sigma}_{t,1} & & \\ & \ddots & \\ & & \hat{\sigma}_{t,r} \end{bmatrix}, \hat{\boldsymbol{\Sigma}}_{t,2} = \begin{bmatrix} \hat{\sigma}_{t,r+1} & & \\ & \ddots & \\ & & \hat{\sigma}_{t,n} \end{bmatrix},$$

$$\sigma_{t,1} \geq \sigma_{t,2} \geq \ldots \geq \sigma_{t,r} \geq \ldots \geq \sigma_{t,d}, \quad \hat{\sigma}_{t,1} \geq \hat{\sigma}_{t,2} \geq \ldots \geq \hat{\sigma}_{t,r} \geq \ldots \geq \hat{\sigma}_{t,d}, \quad r = \sum_{k=1}^K r_k.$$

From Equation (15), we know that $\sigma_{t,r+1} = \ldots = \sigma_{t,d} = 0$. It is easy to show that:

$$\boldsymbol{M} := \hat{V}_{t,1} = \begin{bmatrix} \boldsymbol{M}_{s_1} & \boldsymbol{M}_{s_2} & \ldots & \boldsymbol{M}_{s_K} \end{bmatrix},$$

where $\{s_1, s_2, \ldots, s_K\} = \{1, 2, \ldots, K\}$ satisfied $\omega_{s_1}(\boldsymbol{x}_t) \geq \omega_{s_2}(\boldsymbol{x}_t) \geq \ldots \geq \omega_{s_K}(\boldsymbol{x}_t)$. And $\hat{\sigma}_{t,r} = \sqrt{\alpha_t} \omega_{s_K}(\boldsymbol{x}_t) = \sqrt{\alpha_t} \min_k \omega_k(\boldsymbol{x}_t)$. Based on the Davis-Kahan theorem [84], we have:

$$\begin{aligned}
\| \left( \boldsymbol{I}_d - \boldsymbol{V}_{t,1} \boldsymbol{V}_{t,1}^\top \right) \boldsymbol{M} \|_F &\leq \frac{\|\boldsymbol{\Delta}_t\|_F}{\min_{1 \leq i \leq r, r+1 \leq j \leq d} |\hat{\sigma}_{t,i} - \sigma_{t,j}|} \\
&= \frac{\|\sqrt{\alpha_t} \sum_{k=1}^K \nabla_{\boldsymbol{x}_t} \omega_k(\boldsymbol{x}_t) \boldsymbol{x}_t^\top \boldsymbol{M}_k \boldsymbol{M}_k^\top \|_F}{\sqrt{\alpha_t} \min_k \omega_k(\boldsymbol{x}_t)} \\
&\leq \frac{\sum_{k=1}^K \|\nabla_{\boldsymbol{x}_t} \omega_k(\boldsymbol{x}_t)\|_F \|\boldsymbol{x}_t^\top \boldsymbol{M}_k \boldsymbol{M}_k^\top \|_F}{\min_k \omega_k(\boldsymbol{x}_t)} \\
&= \frac{\alpha_t}{1 - \alpha_t} \frac{C_3 C_4}{\min_k \omega_k(\boldsymbol{x}_t)}
\end{aligned}$$

.

Because $\lim_{t \to 1} \min_k \omega_k(\boldsymbol{x}_t) = \frac{1}{K}$, $\lim_{t \to 1} \frac{\alpha_t}{1 - \alpha_t} = 0$, so:

$$\lim_{t \to 1} \| \left( \boldsymbol{I}_d - \boldsymbol{V}_{t,1} \boldsymbol{V}_{t,1}^\top \right) \boldsymbol{M} \|_F = 0.$$

And from Equation (16), we have:

$$\lim_{t \to 1} \| \left( \boldsymbol{I}_d - \boldsymbol{U}_{t,1} \boldsymbol{U}_{t,1}^\top \right) \boldsymbol{M} \|_F = 0.$$

$\square$

# G   Image Editing and Evaluation Experiment Details

All the experiments can be conducted with a single A40 GPU having 48G memory.

## G.1   Editing in Unconditional Diffusion Models of Different Datasets

**Datasets.** We demonstrate the unconditional editing method in various dataset: FFHQ [63], CelebaA-HQ [52], AFHQ [62], Flowers [61], MetFace [85], and LSUN-church [60].

**Models.**   Following [30], we use DDPM [1] for CelebaA-HQ and LSUN-church, and DDPM trained with P2 weighting [86] for FFHQ, AFHQ, Flowers, and MetFaces. We download the official pre-trained checkpoints of resolution $256 \times 256$, and keep all model parameters frozen. We use the same linear schedule including 100 DDIM inversion steps [3] as [30]. Further, we apply quanlity boosting after $t = 0.2$ as proposed in [87].

**Edit Time Steps.**   We empirically choose the edit time step $t$ for different datasets in the range $[0.5, 0.8]$. In practice, we found time steps within the above range give similar editing results. In most of the experiments, the edit time steps chosen are: $0.5$ for FFHQ, $0.6$ for CelebaA-HQ and LSUN-church, $0.7$ for AFHQ, Flowers, and MetFace.

**Editing Strength.**   In the empirical study of local linearity, we observed that the local linearity is well-preserved even with a strength of $300$. In practice, we choose the edit strength $\lambda$ in the range of $[-15.0, 15.0]$, where a larger $\alpha$ leads to stronger semantic editing and a negative $\alpha$ leads to the change of semantics in the opposite direction.

## G.2   Comparing with Alternative Manifolds and Methods

**Existing Methods**   We compare with four existing methods: NoiseCLR [23], BlendedDiffusion [24], Pullback [30], and Asyrp [29].

**Alternative Manifolds.**   There are two alternative manifolds where similar training-free approaches can be applied, and each of the alternative involves evaluation of the Jacobians $\dfrac{\partial \boldsymbol{\epsilon}_t}{\partial \boldsymbol{h}_t}$ (equivalently $\dfrac{\partial \hat{\boldsymbol{x}}_0}{\partial \boldsymbol{h}_t}$), and $\dfrac{\partial \boldsymbol{\epsilon}_t}{\partial \boldsymbol{x}_t}$ accordingly.

- $\dfrac{\partial \boldsymbol{\epsilon}_t}{\partial \boldsymbol{h}_t}$ (or equivalently $\dfrac{\partial \hat{\boldsymbol{x}}_{0,t}}{\partial \boldsymbol{h}_t}$ up to a scale) calculates the Jacobian of the noise residual $\boldsymbol{\epsilon}_t$ with respect to the bottleneck feature of $\boldsymbol{x}_t$.

- $\dfrac{\partial \boldsymbol{\epsilon}_t}{\partial \boldsymbol{x}_t}$ calculates the Jacobian of the noise residual $\boldsymbol{\epsilon}_t$ with respect to the input $\boldsymbol{x}_t$.

Notably, $\dfrac{\partial \boldsymbol{\epsilon}_t}{\partial \boldsymbol{h}_t}$ has hardly notable editing results on images, and hence we present the editing results of $\dfrac{\partial \boldsymbol{\epsilon}_t}{\partial \boldsymbol{x}_t}$. Besides, with masking and nullspace projection, $\dfrac{\partial \boldsymbol{\epsilon}_t}{\partial \boldsymbol{x}_t}$ also leads to hardly notable changes on images, thus the final comparison is without masking and nullspace projection.

**Evaluation Dataset Setup.**   In human evaluation, for each method, we randomly select 15 editing direction on 15 images. Each direction is transferred to 3 other images along both the negative and positive directions, in total 90 transferability testing cases. Learning time and transfer edit time are averaged over 100 examples. LPIPS [64] and SSIM [65] are calculated over 400 images for each method.

**Human Evaluation Metrics.**   We measure both Local Edit Success Rate and Transfer Success Rate via human evaluation on CelebA-HQ. i) Local Edit Success Rate: The subject will be given the source image with the edited one, if the subject judges only one major feature among {"eyes", "nose", "hair", "skin", "mouth", "views", "Eyebrows"} are edited, the subject will respond a success, otherwise a failure. ii) Transfer Success Rate: The subject will be given the source image with the edited one, and another image with the edited one via transferring the editing direction from the source image. The subject will respond a success if the two edited images have the same features changed, otherwise a failure. We calculate the average success rate among all subjects for both Local Edit Success Rate and Transfer Success Rate. Lastly, we have ensured no harmful contents are generated and presented to the human subjects.

**Learning Time.**   Learning time is a measure of the time it takes to compute local basis(training free approaches), to train an implicit function, or to optimize certain variables that help achieve editing for a specific edit method.

### G.3 Editing in T2I Diffusion Models

**Models.** We generalize our method to three types of T2I diffusion models: DeepFloyd [19], Stable Diffusion [4], and Latent Consistency Model [38]. We download the official checkpoints and keep all model parameters frozen. The same scheduling as that in the unconditional models is applied to DeepFloyd and Stable Diffusion, except that no quality boosting is applied. We follow the original schedule for Latent Consistency Model [38] with the number of inference steps set as $4$.

**Edit Time Steps.** We empirically choose the the edit time step $t$ as $0.75$ for DeepFloyd and $0.7$ for Stable DIffusin. As for Latent Consistency Model, image editing is performed at the second inference step.

**Editing Strength.** For unsupervised image editing, we choose $\lambda \in [-5.0, 5.0]$ in Stable Diffusion, $\lambda \in [-15.0, 15.0]$ in DeepFloyd, and $\lambda \in [-5.0, 5.0]$ in Latent Consistency Model. For text-supervised image editing, we choose $\lambda \in [-10.0, 10.0]$ in Stable Diffusion, $\lambda \in [-50.0, 50.0]$ in DeepFloyd, and $\lambda \in [-10.0, 10.0]$ in Latent Consistency Model.

## H    Social Impacts and Safeguards

The paper originally presents a new image manipulation method, with a theoretical framework to deepen the understanding of diffusion models. However, there exist potential social impacts that the proposed methods can be misused in generating and manipulating harmful content. Therefore, we will release our code and models with license and ethics commitments in the future. Besides, methods for identifying and preventing such harmful behaviors are of great significance in generative models.

