# OpenReview forum: "Exploring Low-Dimensional Subspace in Diffusion Models for Controllable Image Editing"
_NeurIPS.cc/2024/Conference — NeurIPS 2024 poster_

### Official Review · Reviewer_hJrH · 2024-07-12

**Soundness:** 2
**Presentation:** 2
**Contribution:** 2
**Rating:** 5
**Confidence:** 4

**Summary:**

Even though there have been many research papers on conditional Diffusion Models, sampling-based disentangling method remains a challenge. The authors empirically/theoretically show that PMP is locally linear and the singular vectors of the gradient of PMP are in low-rank space.  Based this, the authors propose LOCO Edit that can benefit homogeneity, composability and linearity in semantic controls. Empirical studies are provided to prove the effectiveness of the method.

**Strengths:**

- The proposed idea is theoretically interesting.
- Introduction is well organized.

**Weaknesses:**

1. Literature review of the recent related works
2. Lack of baselines and comparisons with other methods (Please add comparisons with other local/global editing papers.)
3. Uninterpretable semantic changes. For example, in Fig. 5, color, light angle, tower architecture are not controllable. Is it red? Blue? What is the meaning of color? It would be more interesting if the authors can add some interpretable ways to control the semantics (e.g., using label information).
4. Almost everything this paper describes is based on PMP while the actual results are not from PMP but from DDIM.
5. Lack of ablation studies; null space projection
6. The metrics used in Fig. 2 are not intuitive and descriptions are not clear.

**Questions:**

1. (Fig. 2)
- Can the results be generalized? It seems like only a few x_T’s are used, and only one prompt is used (as mentioned in Appendix D.1)
- fig2(a) The authors mentioned in L79 that “rank (A) denotes the numerical rank of A”.  However, in Eq. 10 and Fig. 2,  it is shown as rank ratio. First of all, what is the rank ratio? and how Eq. 10 can be interpreted? Please elaborate them.
- fig2(b) To my understanding, $f$ and $l$ (in L146) seem to be a sort of the denoised images. Why not directly computing their distance? It would be more natural measure for the statement $f \sim l$ in L142.

2. (L80) the x_0 sampled from p_data might need to be the input to get x_t, not the output of the posterior distribution.

3. (L37) What are the various other unexplored aspects?

4. (Fig. 4) What are the undirected t2i edit? If text is not given, why is it called as t2i edit?

5. (L159-165) Are the three properties shown to be true in the experiments? For example, what about composing multiple singular vectors?

**Limitations:**

1. (L157) and (Alg. 1) The authors mentioned that the proposed method benefits the one-step editing while the actual algorithm needs to sample by DDIM twice (DDIM inv and DDIM).

2. (L189-L190) I think there would have been many papers regarding local editing based on Diffusion Models such as Blended diffusion [1].


[1] Blended Diffusion: Text-driven Editing of Natural Images, CVPR’22

---

> ### Author Rebuttal · Authors · 2024-08-07
>
> We appreciate the reviewer's constructive feedback that helps improve the quality of our work. Below, we address the reviewer's major concerns and clarify potential misunderstandings in some parts of our work. And we will incooperate those valuable points into our final version.
> >**Q1:
> >Literature review of related works.
>        (L189-190) I think ... such as Blended diffusion.**
>
> **A1:**
> 1. First, a detailed discussion of related works is provided in Appendix B for space limitations. There, we thoroughly review the existing literature on (1) semantic latent spaces of diffusion models, (2) global image editing in unconditional diffusion models, and (3) local/global image editing in conditional diffusion models.
> 2. Several existing methods address local editing in conditional diffusion models, such as CLIP-guided diffusion in BlendedDiffusion and classifier-free guidance diffusion in SEGA [22]. However, local editing remains challenging for purely unconditional diffusion models. Thanks the reviewer. We will clarify this point in lines 189-190 and include BlendedDiffusion in related works.
>
> >**Q2: Lack of baselines and comparisons...**
>
> **A2:** Thanks to the suggestions, we conduct comparisons with BlendedDiffusion and NoiseCLR [1], with details in **Q&A 1 of the global response.**
>
> >**Q3: Lack of ablation studies; null space projection**
>
> **A3:** We showed the role of nullspace projection in Fig. 8 in Appendix, and refer the summary of ablation studies to **Q&A 2 in the global response**.
>
> >**Q4: Uninterpretable semantic change…**
>
> **A4: These are great points, but we want to give further clarification:**
> 1. **Unsupervised vs supervised edit.** Unsupervised edit finds edit directions without any label or text prompt such as Pullback [25] and NoiseCLR, while supervised edit utilizes labels or text supervision such as BlendedDiffusion and SEGA [22]. Both are important research directions. Our method mostly focuses on unsupervised edit (Fig. 1 and 5), but can be extended to supervised edit with a target text prompt as shown in Fig. 4(b).
> 2. **Interpretation of semantic changes.**  Even in unsupervised edit, the semantic change directions can be interpreted only after being identified. For example, in Fig. 5, these directions indeed have semantic meanings: colors are changed from white to red, tower architecture are changed from simple to complex. We will make the descriptions more concrete.
>
> >**Q5: Misunderstandings on our paper.**
> >1. The actual results are not from PMP but from DDIM
> >2. (L80) The $x_0$ sampled from $p_{data}$ might need to be the input to get $x_t$, not the output of the posterior distribution
> >3. (L157) The proposed method is not one-step editing...
> >4. (L159-165) Are the three properties...
> >5. What are the undirected t2i edit...
>
> **A5:** We want to clarify these misunderstandings:
> 1. **PMP and DDIM.**    We use PMP at time step $t$ to find the direction $v_p$ and edit $x_t$ as $x_t + \lambda v_p$. In contrast, DDIM-Inv is used to get $x_t$ from $x_0$, and DDIM is to generate the edited image by denoising $x_t + \lambda v_p$. We use both PMP and DDIM, whereas PMP takes the most important role in finding the edit direction $v_p$.
> 2. **Clarification on L80.** $\mathbb{E}_{x_0 \sim p\_{data}(x)}[x_0|x_t]$ is the expectation of $x_0$ given the observed $x_t$ and the prior distribution $p\_{data}(x)$. Its output is indeed **not** $x_0$.
> 3. **One-step editing.** In [25], “one step” means only changing $x_t$ at one timestep $t$ for image editing. In comparison, as mentioned by reviewer 9wnQ, NoiseCLR requires edits across multiple timesteps.
> 4. **Experimental verification of three properties.** These properties have been demonstrated in Fig. 1 (b,c,d) and discussed in L 64 - 69. Particularly, in Fig. 1\(c\), the disentangled editing directions are composable: linear combination of two editing directions result in changes of two attributes simultaneously.
> 5. **Undirected and text-directed T2I edit.**
> 	 Suppose T2I diffusion models generate an image based on text prompt $c_o$.
> 	- **In Undirected T2I edit,** image is edited without additional editing prompt for the target edit direction.
> 	-  **In Directed T2I edit,** an extra editing prompt $c_e$ is provided. So it is called “text-directed T2I edit”.
>
> >**Q6: For Fig. 2:**
> >1. Whether the results be generalized to more data samples and prompts
> >2. Define rank ratio and interpret Eq. 10
> >3. Why use the norm ratio
>
> **A6:** Thanks for raising these points.
> 1. We test 15 more text prompts. As shown in **Fig. F of the global response PDF**.  Results demonstrate the generalizability of more prompts and initial noises.
> 2. The rank ratio is defined in Line 144, which is the ratio between the numerical rank and the ambient dimension $d$. Eq.10 can be interpreted as finding the smallest $r$ such that: compared to the top-$r$ largest singular values of the given Jacobian, the remaining singular values are smaller than a threshold and can be neglected.
> 3. The norm ratio $||l - f||_2/||l|_2$ measures relative distance to better reflect linearlity. If $l$ and $f$ are apart from each other but $||l||_2$ and $||f||_2$ are too small, the absolute norm differences $||l - f||_2$ may be too small to reflect violations of linearity, but the norm ratio can.
>
>
> >**Q7: (L37) What are the various other unexplored aspects?**
>
> **A7:** Specifically, these aspects include but not limited to (1) whether diffusion models have semantic spaces; (2) what features lie in the semantic spaces; (3) theoretically justify those semantic features; (4) utilize those features unsupervised local edit in unconditional diffusion models.  In this work, we looked into all these aspects and proposed LOCO edit that tackles many of the questions raised. We will make the discussion clearer.
>
> [1] Yusuf et al., "Noiseclr: A contrastive learning approach for unsupervised discovery of interpretable directions in diffusion models." CVPR 2024.

---

> > ### Comment · Reviewer_hJrH · 2024-08-14
> >
> > Thank you for clarifying my concerns during the rebuttal.
> >
> > Overall, most of my concerns have been resolved.
> > I will raise my score to 5.
> >
> > One of the remaining issues (which could be important) is:
> > if the semantic direction means white to red, what happens if input image does not have anything related to white and red, e.g., black and blue?

---

> ### Author Response · Authors · 2024-08-13
> **looking forward to your reply**
>
> Dear Reviewer hJrH,
>
> We have worked diligently to resolve your concerns.
>
> As the rebuttal period comes to an end, please do not hesitate to contact us if you have any last-minute questions or require further clarification.
>
> Best regards,
>
> The Authors

---

> ### Author Response · Authors · 2024-08-14
>
> Dear reviewer,
>
> Thanks for the positive response and we appreciate your acknowledgment of our responses in addressing the existing problems.
>
> For the proposed precise point: (1) We have tested transferring edit direction from “white to red” to flowers in other colors including blue, pink, and orange. We saw the colors change to darker blue, darker pink, and darker orange. Considering the effects of transferring, the edit direction can be described as “darker color”, which is semantically meaningful for flowers in general. (2) To further discuss the point, when transferring “enlarging eyes” to images of a person wearing sunglasses, only the details of the sunglasses are changed which is hard to notice. This is because “enlarging eyes” is not semantically meaningful for people wearing sunglasses. (3) Such edge cases are interesting, but since we consider practical editing scenarios where the edit directions are semantically meaningful, we do not present them in the paper. Besides, we can not show additional figures, but we will add those in our revision to make the discussion more complete.
>
> Thanks for the question and hope we have further addressed your concerns. We are happy to answer any other questions.
>
> Best Regards,
>
> Authors

---

### Official Review · Reviewer_9wnQ · 2024-07-12

**Soundness:** 4
**Presentation:** 3
**Contribution:** 3
**Rating:** 7
**Confidence:** 4

**Summary:**

This paper proposes an diffusion image editing framework called LOw-rank COntrollable edit (LOCO Edit) based on two observations: 1)  the learned posterior mean predictor (PMP) is locally linear during the middle timesteps during denoising, and 2) the singular vectors of the PMP's Jacobian lie in low-dimensional semantic subspaces. These observations are backed both empirically and theoretically. To conduct LOCO edit, an image is inverted with DDIM inversion to an intermediate timestep. SVD is conducted on the Jacobian of the PMP w.r.t. to this noisy image in order to discover semantic directions. Furthermore, a mask can be applied along with a null space projection in order to achieve more disentangled and localized edits. A variant of LOCO edit is also proposed to enable text direction. A variety of experiments are conducted for the unconditional LOCO edit on various domains using human evaluation and CLIP score and edit discovery time, achieving superior results against other methods. Qualitative results are shown for various diffusion model variants (e.g., Stable Diffusion, Latent Consistency Model) and domains, demonstrating linearity and the ability to modulate strength, the ability to compose with other directions, as well as the ability to transfer the edits to other images.

**Strengths:**

* The paper tackles an important problem in diffusion models: disentangled and continuous editing.

* This is done in a principled way, based on two important observations of the PMP and its Jacobian across different timesteps. Furthermore, there are both empirical experiments as well as theoretical analysis validating these.

* These edits can be composed, transferred, and modulated in a continuous manner. Additionally, the time required for discovering these directions is less than other methods, such as Asyrp.

* The qualitative and quantitative results demonstrate strong editing capabilities.

* A variety of ablations are provided in the paper and the appendix, such as the effect of nullspace projection, timestep used for editing, etc.

**Weaknesses:**

* The qualitative results for editing are not quite the best I have seen, and they are compared against relatively weak baselines (e.g. Asyrp). However, I find that this is okay since the approach is quite principled.  Many impressive diffusion based image editing papers have been released in the past year [1,2]. I don't think it is necessary to compare against them, but including them in the paper in the related works section would keep this paper up-to-date with the current state of image editing.

* Since the linearity and low-rankness properties are more apparent in the middle time-steps, applying more global edits (e.g. changing the shape of something while keeping appearance) would be difficult since more coarse level features are constructed in the earlier timesteps of denoising. I see this effect in Fig. 7 in the appendix, as other parts of the image are edited as well.

[1] @inproceedings{dalva2024noiseclr,
  title={Noiseclr: A contrastive learning approach for unsupervised discovery of interpretable directions in diffusion models},
  author={Dalva, Yusuf and Yanardag, Pinar},
  booktitle={Proceedings of the IEEE/CVF Conference on Computer Vision and Pattern Recognition},
  pages={24209--24218},
  year={2024}
}
[2] @article{gandikota2023concept,
  title={Concept sliders: Lora adaptors for precise control in diffusion models},
  author={Gandikota, Rohit and Materzynska, Joanna and Zhou, Tingrui and Torralba, Antonio and Bau, David},
  journal={arXiv preprint arXiv:2311.12092},
  year={2023}
}

**Questions:**

* I am quite surprised that an edit at a single timestep has enough effect to edit the image. For instance, [1,2] require over multiple timesteps. Is there an intuition for why this edit at a single timestep has enough causal effect?

* For more open-domain diffusion models like Stable Diffusion, can the edit direction be applied to another subject? For instance, in Figure 1, you transfer human opening eyes to different images. Would this transfer to other domains, such as an animal?

* A visualization of all the edit directions discovered via SVD would be quite interesting. Are they all semantically meaningful?

**Limitations:**

A more substantive discussions of limitations is missing. Although Appendix Section H discusses future directions, there is little mention of limitations of the current method.

---

> ### Author Rebuttal · Authors · 2024-08-07
>
> We thank the reviewer for constructive feedback and interesting questions. During rebuttal, we address the reviewer’s concerns on the experiments and other questions as follows.
>
> > **Q1: The qualitative resultes are not quite the best... up-to-date with the current state of image editing.**
>
> **A1:** Thank the reviewer for pointing out the interesting works NoiseCLR and Concept sliders. We have looked into them and will cite and add discussions on NoiseCLR and Concept sliders in the revised manuscript. Besides, based on the reviewer's suggestion, we conducted more qualitative and quantitative baseline comparisons with NoiseCLR. Moreover, we extend both qualitative and quantitative results per the reviewer's suggestion. We refer detailed results and discussion to **Q&A 1 of the global response**.
>
> >**Q2: Since the linearity and low-rankness properties are more apparent in the middle time-steps, applying more global edits... would be difficult since more coarse level features are constructed in the earlier timesteps of denoising...are edited as well.**
>
> **A2:** Thanks for raising this point here, and we would like to discuss further.
> 1. For coarse features that are controlled in early time steps close to random noise, LOCO is not guaranteed to disentangle these features in the high-rank space. This is closely related to our method is principled in the low-rank subspace of diffusion models in the middle time steps, and hence we mainly focus on unsupervised precise local edit.
> 2. It would be interesting to non-trivially model the space at those high-rank timepoints, maybe taken inspiration from works such as NoiseCLR. Our current focus is to study the low-rank subspaces in diffusion models and leave the understanding of semantics in high-rank spaces to be explored in future studies.
>
> > **Q3: I am quite surprised that an edit at a single timestep has enough effect to edit the image. For instance, [1,2] require over multiple timesteps. Is there an intuition for why this edit at a single timestep has enough causal effect?**
>
> **A3:** Thanks for the interesting question and we try to provide possible intuitions.
> 1. From the perspective of method principles, LOCO initiates from the local linearity of PMP, and tries to find directions that lead to the largest changes in the posterior mean. Under the assumption of local linearity within the low-rank semantic subspace, such identified direction $v$ can achieve one step edit with appropriate edit strength. Following the intuition above, the one-step edit ability is experimentally verified.
> 2. In contrast, NoiseCLR optimizes a conditional variable representing specific features to be used in classifier-free guidance, and Concept sliders use LoRA to finetune a Slider that steers the generation to be more aligned with using conditional variable $c+$ instead of $c-$. Such indirect optimization and finetuning based on conditional variables may benefit from multiple timesteps to achieve faster convergence in optimization and better performance in editing.
>
> > **Q4: For more open-domain diffusion models like Stable Diffusion, can the edit direction be applied to another subject? For instance, in Figure 1, you transfer human opening eyes to different images. Would this transfer to other domains, such as an animal?**
>
> **A4:** Thanks for the interesting question, and we would like to discuss it further.
> 1. The transferability has a high success rate for unconditional diffusion, which matches our theoretical analysis for unconditional diffusion models. However, it’s currently hard to transfer edit directions for more open-domain T2I diffusion models as we have experimented.
> 2. It is potentially because the feature space of these diffusion models is more complicated correlating with various text prompts. Such feature space does not align with the assumptions in the theoretical analysis for unconditional diffusion models.
> 3. The modeling of feature space in T2I diffusion models is still an open and interesting question in the area. The exploration of transferability in these models requires further studies that may involve conditional text variables, noisy images, and various other non-trivial aspects.
>
> > **Q5: A visualization of all the edit directions discovered via SVD would be quite interesting. Are they all semantically meaningful?**
>
> **A5:** Thanks for proposing the interesting question. We have attached the identified editing directions for different regions of interest in **Figure C of the global response PDF**. These editing directions are selected from CelebA, FFHQ, and AFHQ.
> 1. We observe that the editing directions demonstrate semantics correspondences to the region of interest. We also notice for these datasets, the position of objects is biased in the center of images, which benefits the transferability of editing directions.
> 2. However, from our observation, the transferability is robust to gender difference, facial feature shape difference, and moderate position difference, as presented in Figure 1(b) of the main paper and **Figure B of the global response PDF**.

---

> > ### Comment · Reviewer_9wnQ · 2024-08-09
> >
> > Thank you for the detailed response. I have read through the global rebuttal and PDF, as well as other responses. Overall, I am satisfied with the responses. I suggest that the authors include the limitations of global editing and the difficulty of transferring edits in conditional diffusion models. After going through this paper again and revisiting other papers, I think there are some similarities to [1], such as linear properties of intermediate noise space, and the one-step editing. I suggest that the authors include this paper in their work and discuss it. I keep my overall score as is.
> >
> > [1] "Boundary Guided Learning Free Semantic Control with Diffusion Models." Zhu et al. 2023.

---

> ### Author Response · Authors · 2024-08-12
>
> Thank you for all the valuable suggestions and for recognizing the value of our work! We will expand the discussion on global editing and transferability in conditional diffusion models and will discuss BoundaryDiffusion in the paper. We have carefully read through BoundaryDiffusion, which is an interesting supervised one-step global edit method. BoundaryDiffusion uses linear SVMs to classify image latent by hyperplanes using label annotations, and about 100 images are required for each class. The edit intuitively moves the image latent in both $\epsilon_t$ and $h_t$ spaces to the negative or positive side of the hyperplane, so that the corresponding semantic is reduced or enhanced. Although the linear and one-step edit properties are similar, our method is (a) localized, (b) requires no label annotation, (c) can find transferable directions using a single image, (d) requires edits only in $x_t$ space, and (e) has a theoretical basis to support the approach. Thanks for the constructive feedback, we will make sure all of them are reflected in our final manuscript.

---

### Official Review · Reviewer_pgJR · 2024-07-13

**Soundness:** 3
**Presentation:** 3
**Contribution:** 3
**Rating:** 5
**Confidence:** 3

**Summary:**

The paper examines the use of low-dimensional subspaces in diffusion models for precise and disentangled image editing. The paper observes that the Posterior Mean Predictor (PMP) in diffusion models shows local linearity across various noise levels, and the singular vectors of its Jacobian exist in low-dimensional semantic subspaces. Building on these observations, they introduce LOw-rank COntrollable image editing (LOCO Edit), a technique that enables efficient, training-free, and precise localized image manipulation by utilizing the Jacobian's low-rank nature. This approach has a strong theoretical basis and has been shown to be effective across different architectures and datasets.

**Strengths:**

1. The paper introduces an approach to image editing within diffusion models by identifying and exploiting low-dimensional subspaces. The idea of using local linearity and low-rank properties of the Jacobian in diffusion models for controllable editing sounds novel.

2. The methodology looks sound, supported by robust empirical evidence demonstrating local linearity and low-rankness of the PMP's Jacobian.

3. The proposed method tackles a key challenge in diffusion models: achieving precise and disentangled image editing without additional training. This has wide-ranging implications for various image generation and manipulation applications.

4. Extensive empirical evaluations are presented, showcasing the method's effectiveness across diverse network architectures (UNet and Transformers) and datasets (e.g., CIFAR-10, CelebA, ImageNet). The results show performance improvements and validate the approach's generalizability.

5. The paper is well-organized and clearly written. Explanations are thorough, and the visuals (e.g., figures) effectively convey the main concepts and outcomes.

**Weaknesses:**

1. The experimental evidence appears highly limited. The quantitative and qualitative comparison results (confined to Table 1 and Fig. 6 with just one example in the main paper) seem inadequate. The study would be strengthened by a more comprehensive comparison with additional state-of-the-art image editing methods, beyond just [24] and [25]. Highlighting specific advantages and potential weaknesses in relation to these methods would provide a more balanced perspective.

2. The assumptions regarding low-rank Gaussian distributions used for theoretical validation warrant more rigorous examination. Expanding the discussion to include potential limitations or circumstances where these assumptions may not hold would add depth and nuance to the analysis.

**Questions:**

1. The paper should clarify that the Posterior Mean Predictor (PMP) is a network function within diffusion models, and detail how it is computed. Specifically, it should describe how PMP is derived and utilized in the context of local linearity and low-rankness.

2. Elaborate on what P_{\omega} and P_{\omega^C} represent in Fig.3 and lines 193-195, and clarify the decomposition of x_{0,t}^{\hat} into ROI and null space using masking techniques.

3. Explain how performing the Jacobian over different masked images helps in disentangling features for precise local editing.

**Limitations:**

In addition to the listed weakness above, the sensitivity of the method to various parameters (e.g., noise levels, perturbation strengths) could be explored in greater detail. Understanding the robustness of the approach under different conditions would be valuable for practical applications.

---

> ### Author Rebuttal · Authors · 2024-08-07
>
> We thank the reviewer for constructive feedback. During rebuttal, we address the reviewer’s concerns on the experiments and other questions as follows.
> >**Q1. Limited experimental evidence and quantitative and qualitative comparisons.**
>
> **A1:** Thanks to the reviewer's suggestions, we added more results, and conducted more qualitative and quantitative comparisons with more baselines to better evaluate our method. We refer detailed results and discussion to **Q&A 1 of the global response**.
>
> >**Q2. Detailed ablation study on sensitivity to various parameters...**
>
> A2: Thanks to the reviewer's suggestions, we conducted more detailed ablation studies and gave a more comprehensive summary. We refer detailed results and discussion to **Q&A 2 of the global response**.
>
> > **Q3: Validation of using the low-rank Gaussian distributions.**
>
> **A3:** In [1], the authors conducted extensive empirical experiments and found that for diffusion models trained on the FFHQ and AFHQ datasets, the learned score can be approximated by the linear score of a Gaussian, particularly at high noise levels. Furthermore, [2] confirms the low-dimensional nature of various image datasets, such as CIFAR-10 and ImageNet, and calculates their intrinsic dimensions. Building on these findings, we incorporate the concept of Gaussian distribution and low-dimensional properties to study the low-rank Gaussian case. This approach enables our model to effectively capture the core structure of image data while remaining tractable for theoretical analysis, thus serving as a practical foundation for theoretical studies on real-world image datasets.
>
> We will integrate these discussions into the final version to enhance motivation.
>
>
> > **Q4. Clarification on Posterior Mean Predictor (PMP).**
>
> **A4:**
> 1.  **Definition & computation of PMP and reference for it’s derivation:** Indeed, PMP is a network function in diffusion models, which takes an input pair ($x_t$, $t$), and outputs the predicted posterior mean $\hat{x}\_{0,t}$. Specifically, the computation of PMP in diffusion models is defined in Equation (2), where $\epsilon_{\theta}$ is the learned unet denoiser in the diffusion model. We refer how PMP is derived to the derivation of Equation (12) in the DDIM paper [3].
> 2.  **How local linearity and low-rankness in PMP is utilized:** (1) PMP’s local linearity is the key assumption for finding edit direction via SVD of PMP’s Jacobian, since directions found via SVD are meaningful only if PMP is linear; (2) PMP’s local linearity also leads to the linear and composable properties of the LOCO edit method; (3) PMP’s low-rankness ensures us to use a low-rank estimation to find the nullspace, achieving efficient and effective nullspace projection.
>
> We will make the above points clear in the final version.
>
> > **Q5: Clarification on finding the precise and disentangled edit direction.**
> > - Elaborate on what $P_{\Omega}$... using masking techniques.
> > - Explain how performing the Jacobian... helps... for precise local editing
>
> **A5:** We would like to clarify concepts, process, and intuition in finding such local edit direction. We will also revise the writing to make the points clearer.
>
> 1.  **Definition of ROI, $P_{\Omega}$ and $P_{\Omega^C}$:** Here, $\Omega$ is an index set that covers the region of interest (ROI) and  $\Omega^C$ covers any other region in the image outside ROI. Based upon this, $P_{\Omega}$ and $P_{\Omega^C}$ denote the projections onto $\Omega$ and $\Omega^C$, respectively. Intuitively speaking, $P_{\Omega}(I)$ crops the content of an image $I$ within the mask, and $P_{\Omega^C}(I)$ crops the contents of $I$ outside of the mask.
> 2.  **Decomposition of $\hat{x}\_{0,t}$ and calculation of Jacobians:** Based on the above definition, we decompose PMP’s output $\hat{x}\_{0,t}$ into $\tilde{x}\_{0,t}$ and $\bar{x}\_{0,t}$ as visualized in Figure 3, where $\tilde{x}\_{0,t} = P_{\Omega}(\hat{x}\_{0,t})$ and $\bar{x}\_{0,t} = P_{\Omega^C}(\hat{x}\_{0,t})$. We further define $\tilde{J}\_{\theta,t} = \partial \tilde{x}\_{0,t} /\partial x_t$ and $\bar{J}\_{\theta,t} = \partial \bar{x}\_{0,t} /\partial x_t$.
> 3.  **Finding image editing directions and nullspace via SVD of Jacobians:** Let $\tilde{J}\_{\theta,t} = \tilde{U}\tilde{S}\tilde{V}^T$, and $\bar{J}\_{\theta,t} = \bar{U}\bar{S}\bar{V}^T$ be the compact SVD. Intuitively, $span(\tilde{V}) = range(\tilde{J}\_{\theta,t}^T)$ is the subspace containing change directions of $x_t$ that leads to change within the $\tilde{x}\_{0,t}$. Besides, $span(\bar{V}) = range(\bar{J}\_{\theta,t}^T)$ is the subspace containing change directions of $x_t$ that leads to edit in $\bar{x}\_{0,t}$. Moreover, $nullspace(\bar{J}\_{\theta,t})$ is the subspace leads to no edit in $\bar{x}\_{0,t}$.
> 4.  **Nullspace projection for more precise and disentangled image editing:** For a direction $v \in range(\tilde{J}\_{\theta,t}^T)$, nullspace projection means projecting $v$ onto $nullspace(\bar{J}\_{\theta,t})$. In practice, this nullspace projection can be calculated as $v_p = (I - \bar{V}\bar{V}^T) v$. Such nullspace projection can further eliminate the effect of $v_p$ to change within $\bar{x}\_{0,t}$, and disentangle the change to be only within $\tilde{x}\_{0,t}$. Then by denoising $x_t + \lambda v_p$, the new image is more precisely edited within the ROI $\Omega$.
>
> [1] Binxu et al., "The Hidden Linear Structure in Score-Based Models and its Application." 2023.
>
> [2] Phillip et al., "The intrinsic dimension of images and its impact on learning." ICLR 2021.
>
> [3] Jiaming et al., Denoising diffusion implicit models. 2020.

---

> > ### Comment · Reviewer_pgJR · 2024-08-14
> >
> > I appreciate the authors' efforts in addressing concerns through their rebuttal. While some issues have been clarified, I still have reservations about the depth of empirical evidence presented in the original paper:
> >
> > 1. Limited comparative analysis: The paper only includes two comparison methods ([24, 25]), which may not provide a comprehensive benchmark.
> >
> > 2. Small sample sizes: The evaluation relies on relatively few samples (90 cases for transferability testing, 100 examples for other metrics), potentially limiting the robustness of the findings.
> >
> > 3. Absence of standard metrics: Apart from using the CLIP score, the study doesn't utilize other widely accepted measures such as Fréchet Inception Distance and Inception Score, which could have strengthened the evaluation.
> >
> > 4. Insufficient ablation studies: The paper lacks thorough ablation studies to demonstrate the contribution of individual components.
> >
> > While the rebuttal has partially addressed these concerns, the original paper's limitations in empirical validation remain significant. Consequently, I am inclined to either maintain my initial score or consider a slight downward adjustment.

---

> ### Author Response · Authors · 2024-08-14
>
> Thanks for the questions, we would like to further make clarifications on these points.
>
> 1. We have extended the comparisons with **two additional baselines** BlendedDiffusion [2] and NoiseCLR [1] for comparison. See global rebuttal Q&A 1 and pdf Tab. A and Fig. E.
>
> 2. (1) The human evaluation dataset is randomly selected have high diversity to cover various semantic directions; (2) Asyrp [3] uses only 40 data samples for human evaluation on CelebA (Appendix K.1), Pullback [4] only shows qualitative results; (3) BlendedDiffusion and NoiseCLR do not mention the size of evaluation dataset; (4) A further detail is, we used 400 samples for other added metrics LPIPS and SSIM. Hence, we think the evaluation dataset size is robust enough to provide fair comparisons.
>
> 3. The mentioned FID and IS are good metrics measuring whether generated image distributions are similar to the training images, but **not standard metrics for image editing methods**. (1) FID and IS will be increased if additional features are edited in comparison to the original one; (2) Representative baslines including NoiseCLR, BlendedDiffusion, Asyrp, and Pullback use neither of the FID and IS; (3) As in global rebuttal Q&A 1 and pdf Tab. A, we have conducted a comprehensive comparison using 7 metrics and 4 attributes to show the superiority of our method. This indeed supports that we have conducted comprehensive empirical evidence in addition to the solid theoretical basis which is lacking in the previous papers.
>
> 4. For more ablation studies, as summarized in Q2&A2 for the global response, we do more ablation studies on Noise levels (i.e., time steps), Perturbation strength, Nullspace projection, and ranks. We have shown representative examples, and more results will be included in the revision.
>
> We appreciate the reviewers' prompt responses and make further clarification and references on how we conduct comprehensive experiments to show the superiority of our method. We kindly request the reviewer evaluate our work fairly based on the rebuttal and our further clarifications.
>
> [1] Yusuf et al., Noiseclr: A contrastive learning approach for unsupervised discovery of interpretable directions in diffusion models. CVPR 2024.
>
> [2] Omri et al., Blended diffusion for text-driven editing of natural images. CVPR 2022.
>
> [3] Mingi et al. Diffusion models already have a semantic latent space. ICLR 2023.
>
> [4] Yong et al. Understanding the latent space of diffusion models through the lens of riemannian geometry. NeurIPS 2023.

---

### Official Review · Reviewer_h6gb · 2024-07-13

**Soundness:** 3
**Presentation:** 3
**Contribution:** 2
**Rating:** 5
**Confidence:** 4

**Summary:**

The paper presents a method for steering the generation in a diffusion model, without any further training. The proposed method is based on two insights about the Posterior Mean Predictor (PMP) and its Jacobian -- that is, the former being locally linear and the latter having singular vectors lying on low-dimensional subspaces. Some theoretical results are provided towards the justification of the above. Local linearity of the PMP allows for a single-step, training-free method for local editing of regions of interest, while the low-rank nature allows for the effective identification of semantic directions using subspace power methods. Some qualitative and quantitative experimental results are provided.

**Strengths:**

The paper is well-written and sound. The provided theoretical results wrt local linearity of the PMP, along with the low-rankness of its Jacobian are interesting (however not extremely surprising) and might be useful to the research community.

**Weaknesses:**

The weakest aspect of the paper concern the reported experimental results. Both qualitative and quantitative results are extremely limited, whilst comparisons with existing work are also not convincing. Besides, the significance of the empirical results is not convincing. For instance, in Fig. 6, when the proposed method (LOCO) is compared to Asyrp, it's not clear at all why localized editing with LOCO is better than Asyrp. Similarly, in Table 1, the Transfer CLIP Score (an important metric in the context of the task studied in the paper) of Asyrp is significantly better that the proposed method's. The authors do discuss this, but the fact that the proposed framework is learning-free, in contrast to Asyrp that learns how to perform editing, does not justify the significant difference wrt to CLIP Score.

Finally, as the authors acknowledge in the Appendix H ("Future Direction", where limitations are also discussed), the provided theoretical framework concerns mainly the undirected image editing part; how text-directed image editing behaves is not addressed by the proposed method. This is a certain limitation, that weakens the generality of the proposed method, yet it doesn't reduce the importance of the theoretical results provided.

**Questions:**

Please see weaknesses.

**Limitations:**

The authors discuss adequately the limitations of the proposed method in the Appendix H (Future Direction). A separate "Limitations" section is missing, but the authors discuss the limitations of the paper honestly and comprehensively.

---

> ### Author Rebuttal · Authors · 2024-08-07
>
> We thank the reviewer’s constructive feedback, and in the following we address the reviewer’s concerns one-by-one.
>
> >**Q1: Not enough qualitative and quantitative results and comparisons with existing work.**
>
> Thanks to the reviewer's suggestions, we conduct more qualitative and quantitative comparisons with more baselines, and add more results to better evaluate our method. We refer detailed results and discussion to **Q&A 1 of the global response.**
>
> >**Q2. Not convincing empirical results.**
> >1. Fig. 6 is not convincing to show better localized editing with LOCO than Asyrp
> >2. Why Transfer CLIP Score of Asyrp is better than LOCO, and whether it’s an issue
>
>
> **A2:** We thank the reviewer for raising these constructive points which improve our work.
> 1. **More convincing visualizations.**
> 	- To demonstrate the effectiveness of our method, we selected examples in Fig. 6 where Asyrp achieves its best performance, yet our method still performs even better. For instance, in Fig. 6, to edit lips, Asyrp noticeably alters undesired regions such as face color, hair shape, and face shape more than our method.
> 	- There are many other instances where Asyrp's editing changes the image much more significantly than ours. We add more random examples in **Figure E of the global PDF** to visualize the local edit ability of our method in comparison of others.
> 2. **Discussion on the Transfer CLIP Score in Table 1\.** **Very good question for a deeper discussion.**
>    - The higher transfer CLIP score of Asyrp is because it is directly supervised by the CLIP score to learn each concept and predict editing directions. In contrast, our method is unsupervised and learning-free, and has the best CLIP score among other unsupervised methods.
>    - Moreover, the transfer CLIP score is biased by large changes in Asyrp: For successful edit in both Asyrp and LOCO, Asyrp tends to have larger global changes resulting in a much higher transfer CLIP score, sometimes leading to corruption in other parts though the desired region is edited. Examples are shown in **Figure E of the global PDF**.
>    - Additionally, the transfer CLIP score relies on CLIP's intrinsic failures to capture detailed semantics as discovered in [1], which may lead to edit failures of Asyrp. Failure examples that are potentially related are shown in **Figure E (row 5-8) of the global PDF**, where Asyrp fails to edit darker eyebrows for all random examples.
>    - Therefore, to compensate for the biases of the transfer CLIP score, we also measure the local edit success rate, where our method outperformed all others. Thanks to the reviewer for raising the point, we also add LPIPS [2] and SSIM [3] as guardians of dramatic changes and use the less biased local edit success rate as the major evaluation metric for local edit ability.
>
> In the revision, we will include those discussions in the paper to clarify our findings.
>
> >**Q3: The provided theoretical framework concerns mainly the undirected image editing part; how text-directed image editing behaves is not addressed by the proposed method. This is a certain limitation, that weakens the generality of the proposed method, yet it doesn't reduce the importance of the theoretical results provided.**
>
> **A3:** We thank the reviewer for raising the points, but we want to clarify this further.
> 1. In Figure 4, we showed that our method can be extended to text-directed image editing. This demonstrates the generality of our edit approach.
> 2. Second, the empirical observation of low-rankness and local linearity is generalizable to text-to-image diffusion models as shown in Figure 2 and Figure 9.
> 3. Although our theoretical study part is for unconditional diffusion models, to the best of our knowledge, our work is the first theoretically ground method compared to all previous diffusion model based editing methods. It's an advantage compared to other image edit methods.
> 4. Besides, for studying text-directed diffusion models, the challenges lie in the more complicated feature space caused by a mixture of image and text distributions. Understanding their feature space is yet an unsolved problem in the area. Hope our work can provide some inspiration for future exploration.
>
> [1] Shengbang et al., Eyes wide shut? exploring the visual shortcomings of multimodal llms. CVPR 2024
>
> [2] Richard et al. The unreasonable effectiveness of deep features as a perceptual metric. CVPR 2018.
>
> [3] Zhou et al., Image Quality Assessment: From Error Visibility to Structural Similarity. 2004

---

### Official Review · Reviewer_eLFT · 2024-07-22

**Soundness:** 3
**Presentation:** 3
**Contribution:** 3
**Rating:** 6
**Confidence:** 4

**Summary:**

This paper made an interesting observation on local linearity of the diffusion model's denoiser. Based on the observation, the author introduces a novel method of performing one-step closed-forrm operation to achieve semantic image editing. Empirical and numerical results are given to demonstrate the effectiveness of the method

**Strengths:**

1. The observations in terms of local linearity of the denoiser is interesting, and is well-validated across different architecture and datasets.
2. The presented method is novel and well-motivated, and simple.
3. The discovery of homogeneity and transferability of the editing direction is insightful.
4. The empirical and numerical result demonstrate consistent image editing.

**Weaknesses:**

1. The computation time of the method is still a bit long (taking around 70s if I understand correctly).

**Questions:**

1. I wonder whether the homogeneity of the editing can be observe on samples that are very different: e.g., 1) computing editing direction on an image with face at the left, and transfer to an image with face on the right; 2) computing editing direction on an image with realistic style, and transfer to an image with cartoon style.
2. In addition, since current models such as SD3 are using flow-matching objective to train the model, I wonder if such observed phenomenon would be more prominent on these models?

**Limitations:**

Yes, the limitations have been adequately addressed.

---

> ### Author Rebuttal · Authors · 2024-08-07
>
> We thank the reviewer’s constructive feedback, and in the following we address the reviewer’s concerns one-by-one.
> >**Q1: The computation time of the method is still a bit long (taking around 70s if I understand correctly)**
>
> **A1:**
> 1. Compared with the global edit method Pullback [25], the additional cost is introduced from finding local edit direction, yet the addition cost is not significant.
> 2. Besides, we add additional comparisons with a local edit method BlendedDiffusion [1] in Table A of the global response PDF, where our method is more efficient in learning time.
> 3. Moreover, our method can transfer the local edit direction to other images with a high successful rate and no additional time cost in learning. On the contrary, other local/global edit methods cannot transfer the edit direction (BlendedDiffusion) or have weaker transferability as summarized in Table A.
>
> >**Q2: Whether the homogeneity of the editing can be observed on samples that are very different:** e.g.,
> >- computing editing direction on an image with a face at the left, and transfer to an image with a face on the right;
> >- computing editing direction on an image with realistic style, and transfer to an image with cartoon style.
>
> **A2:**
> This is a good question. We present more results and extend the discussions.
> 1. - For unconditional diffusion models on FFHQ and CelebA, we can transfer edit directions to faces on the different side, as presented in **Figure B of the global response PDF**. But we do notice that these datasets tend to have the object in the center, which benefits the transferability.
> 	- We have further attached the visualization of editing directions in Figure C, and the edit direction has semantic correspondences to the target edit region. From our observation, the transferability is robust to gender differences, facial feature shape differences, and moderate position differences in FFHQ and CelebA.
> 2. - To achieve goal (2), conditional diffusion models are required. We have previously explored such transferability in stable diffusion models, and the transfer success rate is low. By our analysis, the homogeneity needs the images lie in the same subspace, but the manifold of images with different styles in conditional diffusion models may violate this.
> 	- Understanding the feature space of conditional diffusion models is a challenging and unsolved problem in the area, since the feature space is related to text prompts and noisy images. We are interested in exploring the question, and hope our theoretical analysis in unconditional diffusion models can provide some inspiration for future works.
>
>
> >**Q3: In addition, since current models such as SD3 are using the flow-matching objective to train the model, I wonder if such observed phenomenon would be more prominent on these models?**
>
> **A3:**
> 1. This is an interesting question and we have explored it further. Due to time constraints, we study the low-rankness and local linearity of SiT [2], a simpler diffusion model using the flow-matching objective at training. As presented in **Figure F of the global response PDF**, we do see generalized local linearity, and a similar low-rank trend in SiT, though the rank is not exactly as low as other diffusion models. It would be interesting to study the differences in low-rank subspace between diffusion models trained under different objectives, and test other diffusion models trained using flow-matching objectives such as Stable Diffusion 3.
>
> [1] Omri et al., Blended diffusion for text-driven editing of natural images. CVPR 2022.
>
> [2] Nanye et al., "Sit: Exploring flow and diffusion-based generative models with scalable interpolant transformers." 2024.

---

### Author Rebuttal · Authors · 2024-08-06

We thank all reviewers for carefully reviewing our work with constructive and positive feedback. Most reviewers find our empirical observation “interesting”, “well-validated”, “important” (eLFT, h6gb, 9wnQ), our theoretical analysis “useful”, “strong”, (h6gb, pgJR), our edit method “novel”, “quite principled”, “well-motivated”, “insightful” (eLFT, pgJR, 9wnQ, hJrH), and our presentation “well-written”, “sound”, “well-organized” and “clear” (h6gb, pgJR, hJrH).

**Summary of our results.** Our work introduced a simple yet effective method for local edit in diffusion models by exploring (i) the linearity of the posterior mean predictor (PMP), and (ii) low-rankness of its Jacobian. The advantages of our method can be highlighted as follows:
-   **One-step, local editing.** To our knowledge, this is the first work on image editing using unconditional diffusion models that allow for one-step, localized editing. In contrast, most works require editing on multiple (all) timesteps and only perform global edit.
-   **An intuitive and theoretically grounded approach.** Our method is highly interpretable, leveraging the benign properties of PMP. The identified properties are well supported by empirical observation (Fig, 2 in paper) and theoretical justifications in Section 4. In contrast, most previous methods are heuristic and lack theoretical justifications.

**Addressing reviewers’ major concerns.** We appreciate the reviewer's comments on our limited experiments. During the revision, we addressed reviewers’ concerns with more comprehensive results, comparisons, and ablation studies as follows. **They are presented in Fig. A-F and Tab. A of the global response PDF**, and will be included in the revised paper, with code released.
>**Q1: More evaluation of our method.**

**A1:**
 1.  **More qualitative results:** We add more qualitative results on different datasets, shown in Fig. A of the attached PDF.
 2.  **More quantitative metrics:** The quantitative comparison is extended in Tab. A of the attached PDF using the additional metrics:
	 - *LPIPS* [4] and *SSIM* [5] to measure the consistency between edited and original images.
	 - *Learning time*, *Transfer Edit Time* to measure the time each method requires to find the editing direction and apply it to edit an image.
     - *#Images for Learning* to measure the number of images used to find directions.
     - *One-step edit*, *No Additional Supervision*, *Theoretically Grounded*, and *Localized Edit* are different properties for each editing method.
3.  **More qualitative comparisons:** We also extend the qualitative comparison in Fig. E to showcase our method's strong local editing capability.
4.  **Detailed comparison with more baselines:** We compare our work with two other studies: NoiseCLR [1] and BlendedDiffusion [2], together with the previous baselines. We discuss key observations as follows.
	-  **Local edit ability:** Tab. A shows LOCO achieves the best Local Edit Success Rate. For LPIPS and SSIM, our method performs better than global edit methods but worse than BlendedDiffusion. However, BlendedDiffusion sometimes fails the edit within the masks (as visualized in Fig. F, rows 1, 3, 4, and 5). We discuss the potential causes from CLIP bias in the last point below. Other methods like NoiseCLR find semantic direction more globally, such as style and race, leading to worse performance in Local Edit Success Rate, LPIPS, and SSIM for localized edits.

	-  **Efficiency and transferability.** First, LOCO requires less learning time than most other methods, and the learning needs only a single time step and a single image. Moreover, LOCO is highly transferable, having the highest Transfer Success Rate in Tab. A. In contrast, BlendedDiffusion can't transfer and requires optimization for each image. While NoiseCLR excels at open-domain transfer, it performs worse than LOCO in closed-domain transfer (e.g., on the CelebA dataset). Other methods exhibit even weaker transferability.
	-  **Theoretically grounded and supervision-free.** LOCO is theoretically grounded. Besides, it is supervision-free, thus integrating no biases from other modules such as CLIP. [3] shows CLIP sometimes can't capture detailed semantics such as color. We observe failures to capture detailed semantics in methods that utilize CLIP guidance such as BlendedDiffusion and Asyrp in Fig. E.
>**Q2: More ablation studies.**

**A2:**
1.  **Noise levels (i.e., time steps).** We test different noise levels and show results in Fig. 7. Results for more noise levels are in Fig. D with key observations: (a) Edit at large noise level (i.e., large time step) perform coarse changes while small noise level performs finer edit; (b) LOCO applies to a large range of noise levels ([0.2T, 0.7T]) for precise edit.
2.  **Perturbation strength.** The linearity concerning edit strengths is visualized in Fig. 1d, and detailed ablation results of perturbation strength at 0.6T are in Fig. D as an example, with key observations: LOCO applies to a generally wide range of perturbation strengths ([-15, 15]) to achieve localized edit.
3.  **Nullspace projection and ranks.** Ablabtion study on nullspace projection is in Fig. 8, with key observations: (a) the local edit ability with no nullspace projection is weaker than that with nullspace projection; (b) when conducting nullspace projection, an effective low-rank estimation with $r=5$ can achieve good local edit results.

[1] Yusuf et al., Noiseclr: A contrastive learning approach for unsupervised discovery of interpretable directions in diffusion models. CVPR 2024.

[2] Omri et al., Blended diffusion for text-driven editing of natural images. CVPR 2022.

[3] Shengbang et al., Eyes wide shut? exploring the visual shortcomings of multimodal llms. CVPR 2024

[4] Richard et al. The unreasonable effectiveness of deep features as a perceptual metric. CVPR 2018.

[5] Zhou et al., Image Quality Assessment: From Error Visibility to Structural Similarity. 2004

---

### Author Response · Authors · 2024-08-12
**Looking forward to your reply**

Dear Reviewers and ACs,

Thank you again for your comprehensive review and valuable suggestions for improvement. We have carefully considered your constructive feedback, and provided detailed responses. With the discussion period concluding in less than two days, we kindly request you to read our rebuttal and we’d appreciate your prompt responses. We are hopeful that our rebuttal has effectively addressed your concerns and you can reconsider your score. If you need any further clarification, we are willing to provide any additional information to ensure your confidence in making responses.

Sincerely,

Authors

---

### Decision · Program_Chairs · 2024-09-25

**Decision:**

Accept (poster)

**Comment:**

This paper proposes a diffusion image editing framework called LOw-rank COntrollable edit (LOCO Edit) based on two observations: 1) the learned posterior mean predictor (PMP) is locally linear during the middle timesteps during denoising, and 2) the singular vectors of the PMP's Jacobian lie in low-dimensional semantic subspaces. These observations are backed both empirically and theoretically. To conduct LOCO edit, an image is inverted with DDIM inversion to an intermediate timestep. SVD is conducted on the Jacobian of the PMP w.r.t. to this noisy image in order to discover semantic directions. Furthermore, a mask can be applied along with a null space projection in order to achieve more disentangled and localized edits. A variant of LOCO edit is also proposed to enable text direction.

The reviewers found the observations in terms of local linearity of the denoiser interesting and well-validated across architectures and dataset. They found the method novel, well-motivated, and appreciated its support on theoretical arguments, finding the discovery of homogeneity and transferability of the editing direction insightful. They also pointed out that LOCO tackles an important challenge in diffusion models: achieving precise and disentangled image editing without additional training, and found the paper to be well written.

There were some concerns about missing citations and comparisons to previous works and, most importantly, limited experimental validation. While the paper considers various architectures and performs various ablations, there were limited examples of the success of the edits. Various questions of detail were also raised. The authors addressed these issues in their rebuttal, providing several new examples of edits. While these are still only for object centric images, they show a wealth of image modifications that can be of interest for applications. In the end, all reviewers were positive towards the paper, and the ACs consur that the paper merits publication.